# PaTH Attention: Position Encoding via Accumulating Householder Transformations

**Songlin Yang**[1]    **Yikang Shen**[2]    **Kaiyue Wen**[3]    **Shawn Tan**[2]
**Mayank Mishra**[2]    **Liliang Ren**[4]    **Rameswar Panda**[2]    **Yoon Kim**[1]

[1]Massachusetts Institute of Technology    [2]MIT-IBM Watson AI Lab
[3]Stanford University    [4]Microsoft

yangsl66@mit.edu

## Abstract

The attention mechanism is a core primitive in modern large language models (LLMs) and AI more broadly. Since attention by itself is permutation-invariant, position encoding is essential for modeling structured domains such as language. Rotary position encoding (RoPE) has emerged as the de facto standard approach for position encoding and is part of many modern LLMs. However, in RoPE the key/query transformation between two elements in a sequence is only a function of their relative position and otherwise independent of the actual input. This limits the expressivity of RoPE-based transformers. This paper describes PaTH, a flexible data-dependent **p**osition encoding scheme based on **a**ccumulated products of **H**ouseholder(like) **t**ransformations, where each transformation is data-dependent, i.e., a function of the input. We derive an efficient parallel algorithm for training through exploiting a compact representation of products of Householder matrices, and implement a FlashAttention-style blockwise algorithm. Across both targeted synthetic benchmarks and moderate-scale real-world language modeling experiments, we find that PaTH improves upon RoPE and other recent baselines. Finally, we show that we can convert pretrained RoPE transformers into PaTH with continued pretraining.

## 1 Introduction

Attention mechanisms form the backbone of transformer architectures that power contemporary AI systems. Attention is inherently permutation-invariant, and thus encoding positional information into attention is important for effective sequence modeling. Since the original sinusoidal embeddings [77], various position encoding schemes have been proposed over the years [16, 63, 28, 25, 45, 58, 72, *inter alia*]; see Dufter et al. [17] for a comprehensive survey. Among these, rotary position embedding [RoPE; 72] has emerged as the de facto standard, adopted in most recent state-of-the-art LLMs.

RoPE works by transforming the key ($\mathbf{k}_j$) and query ($\mathbf{q}_i$) embeddings through a rotation matrix $\mathbf{R}$ whose rotation angle is a function of the difference in positions, resulting in the bilinear form $\mathbf{q}_i^\top \mathbf{R}^{i-j} \mathbf{k}_j$ for the attention logits. The rotation matrix $\mathbf{R}$ itself is a block-diagonal matrix composed of two-by-two rotation matrices, which enables efficient computation. However, the rotation matrix in RoPE is *data-independent* and only a function of the relative position (i.e., $\mathbf{R}$ applied $i - j$ times), which limits its expressivity; indeed, recent work [7] demonstrates that RoPE-based transformers are still computationally constrained to the $\mathsf{TC}^0$ complexity class, the complexity class of ordinary transformers with absolute position embeddings [49]. As a potential consequence, RoPE-based transformers have been empirically found to have difficulty with simple synthetic tasks that require a form of sequential reasoning, such as flip-flop language modeling [41] and certain state-tracking tasks [51]. Insofar as such simple sequential reasoning underlie real-world capabilities that we want

---

The implementation of the PaTH attention layer is also made available as part of the FLASHLINEARATTENTION library [80, 79]: https://github.com/fla-org/flash-linear-attention

in our LLMs, these failure modes highlight the need to design new primitives that can overcome these theoretical and empirical limitations of existing attention layers.

This work develops PaTH, a **p**osition encoding scheme with **a**ccumulated **H**ouseholder **t**ransformations, targeting the above problem. In PaTH, the attention logit is still parameterized as a bilinear form $\mathbf{q}_i^\top \mathbf{H}_{ij} \mathbf{k}_j$, but the matrix $\mathbf{H}_{ij} \in \mathbb{R}^{d \times d}$ is obtained via a cumulative product of *data-dependent* matrices along the path between positions $j$ and $i$, where the matrices have Householder-like identity-plus-rank-one structure. Intuitively, this formulation captures the cumulative transformation between positions, enabling PaTH to dynamically adapt to input data and solve certain state-tracking problems. Indeed, we show that a constant-layer PaTH-based transformer can solve an $\mathsf{NC}^1$-complete problem under $\mathsf{AC}^0$ reductions, i.e., PaTH can extend transformers beyond the $\mathsf{TC}^0$ complexity class (assuming $\mathsf{TC}^0 \neq \mathsf{NC}^1$).

To scale up PaTH Attention, we develop a FlashAttention-like algorithm [14] for hardware-efficient parallel training that leverages a compact representation of products of Householder matrices [5, 27]. Empirical results show that PaTH-based models can solve challenging synthetic state-tracking tasks where RoPE-based Transformers struggle. On moderate-scale language modeling with 760M-parameter Transformers, PaTH outperforms both RoPE and the Forgetting Transformer [39], which modulates attention logits via a data-dependent additive term. Combining PaTH with the Forgetting Transformer yields further gains, and the resulting models generalize well beyond the training sequence length. Finally, we show that we can convert pretrained RoPE transformers into PaTH with continued pretraining.

## 2 PaTH Attention

PaTH employs a dynamic data-dependent transition matrix—in particular identity-plus-rank-one Householder-like transformations—for computing the bilinear attention logits, unlike RoPE which applies a fixed transformation at each time step.

### 2.1 Generalizing RoPE with Multiplicative Position Encodings

Traditional additive position encodings, such as sinusoidal embeddings [77] or ALiBi [58], represent positions as vectors or matrices summed directly with token embeddings or attention logits. RoPE instead encodes relative positions multiplicatively rather than additively by directly modulating the key/query vectors via position-dependent transformations. The class of multiplicative positional encodings can more generally be defined as $\mathbf{A}_{ij}$ such that,

$$\mathbf{A}_{ij} \propto \exp\Big(\mathbf{k}_j^\top \Big( \prod_{s=j+1}^{i} \mathbf{H}_s \Big) \mathbf{q}_i \Big),$$

where $i$ and $j$ are positions of the query and key, and $\mathbf{H}_s \in \mathbb{R}^{d \times d}$ is a *transition matrix*. RoPE is thus a special case of the above with a static transition matrix $\mathbf{H}_s = \mathbf{R}$, where $\mathbf{R}$ is a block diagonal with $d/2$ independent 2-dimensional rotation blocks, each of which has different rotation angles. This static rotation structure allows for efficient computation of RoPE-based attention in practice.

### 2.2 Data-dependent Multiplicative Position Encodings with PaTH

PaTH employs a *data-dependent* Householder-like[1] matrix with identity-plus rank-one-structure:

$$\mathbf{H}_t = \mathbf{I} - \beta_t \mathbf{w}_t \mathbf{w}_t^T,$$

where $\mathbf{w}_t \in \mathbb{R}^d$ and $\beta_t = 2 \times \mathrm{sigmoid}(\mathbf{u}^\top \mathbf{x}_t + b) \in (0, 2)$ are functions of the current input $\mathbf{x}_t$.[2] We motivate this parameterization from the perspective of generalizing expressive linear RNNs.

Concretely, consider linear attention transformers with matrix-valued hidden states $\mathbf{S}_t \in \mathbb{R}^{d \times d}$ with the above Householder-like transition function, where the output $(\mathbf{o}_t)$ given the key $(\mathbf{k}_t)$, query $(\mathbf{q}_t)$, value $(\mathbf{v}_t)$ vectors is given by

$$\mathbf{S}_t = \mathbf{S}_{t-1}\mathbf{H}_t + \mathbf{v}_t\mathbf{k}_t^\top, \qquad\qquad \mathbf{o}_t = \mathbf{S}_t\mathbf{q}_t.$$

---

[1]Householder matrices take the form $\mathbf{I} - \frac{2}{\|\mathbf{u}\|_2^2}\mathbf{u}\mathbf{u}^\top$ and hence our matrix is only Householder-like.

[2]We use $\beta_t \in (0, 2)$ as this allows for negative eigenvalues in the transition matrix [22], which has been shown to boost the state tracking performance in the DeltaNet case [22, 69]. The vector $\mathbf{w}_t$ is obtained by applying a low-rank linear layer followed by a short convolution layer (filter size 3) and an $L_2$ normalization layer. Hence PaTH only adds a small number of additional parameters.

Recent works have shown that such linear RNNs empirically achieve good performance on language modeling [66, 78, 82]. And despite being more efficient than softmax attention, these models have been shown to be (in a certain way) more expressive than transformers [22, 69], in particular being able to solve a class of *state tracking* problems that cannot be solved by ordinary transformers. Now consider unrolling the recurrence in the RNN, and compare it against the PaTH-attention output,

$$\text{RNN: } \mathbf{o}_t = \sum_{j=1}^{t} \mathbf{v}_j \left( \mathbf{k}_j^\top \left( \prod_{s=j+1}^{t} \mathbf{H}_s \right) \mathbf{q}_t \right), \quad \text{PaTH: } \mathbf{o}_t = \frac{1}{Z_t} \sum_{j=1}^{t} \mathbf{v}_j \exp\left( \mathbf{k}_j^\top \left( \prod_{s=j+1}^{t} \mathbf{H}_s \right) \mathbf{q}_t \right),$$

where $Z_t = \sum_{j=1}^{t} \exp\left( \mathbf{k}_j^\top \left( \prod_{s=j+1}^{t} \mathbf{H}_s \right) \mathbf{q}_t \right)$ is the normalizer. This view shows that PaTH is closely related to such expressive linear RNNs, and we thus expect PaTH-based transformers to inherit their increased expressivity. Indeed, the following theorem shows that PaTH can extend transformers beyond the $\mathsf{TC}^0$ complexity class.

**Theorem 2.1.** *A one-layer PaTH transformer with two attention heads and* $\log n$ *precision can solve an* $\mathsf{NC}^1$*-complete problem under* $\mathsf{AC}^0$*-reductions.*

The proof, given in appendix A, is a straightforward adaptation of Theorem 2 from Peng et al. [56], which showed the that linear RNNs with a similar data-dependent transition matrix can solve an $\mathsf{NC}^1$-complete problem. However, such RNNs still have theoretical limitations that attention does not have, for example in its (in)ability to perform associative recall over a given context of arbitrary length [2]. In contrast, PaTH can capture the benefits of both softmax attention (associative recall) and expressive linear RNNs (state tracking).

**Extension: PaTH-FoX.** PaTH simply provides a more expressive way to encode unnormalized attention logits and is thus compatible with other recently proposed modifications to softmax attention such as Stick-Breaking Attention [73], Selective Attention [35], and Forgetting Transformer [FoX; 39]. As a case study we experiment with combining PaTH with FoX, which *additively* modifies the attention logits in a data-dependent manner. We show that this combined strategy leads to improved performance on some downstream tasks, especially in length extrapolation.

Concretely, FoX [39] modifies the attention via data-dependent "forget" gates $f_s \in (0, 1)$

$$\mathbf{A}_{ij} \propto \exp\left( \mathbf{k}_j^\top \mathbf{q}_i + \sum_{s=j+1}^{i} \log f_s \right) = \left( \prod_{s=j+1}^{i} f_s \right) \exp\left( \mathbf{k}_j^\top \mathbf{q}_i \right),$$

where $f_s = \text{sigmoid}(\mathbf{u}_f^\top \mathbf{x}_s + b_f)$. Similar to how PaTH can be seen as a softmax version of DeltaNet-style linear RNNs [65, 81], FoX can be seen as softmax version of GLA-/Mamba2-style linear RNNs [80, 13].[3] We can combine the two mechanisms to arrive at PaTH-FoX attention:

$$\mathbf{A}_{ij} \propto \left( \prod_{s=j+1}^{i} f_s \right) \exp\left( \mathbf{k}_j^\top \left( \prod_{s=j+1}^{i} \mathbf{H}_s \right) \mathbf{q}_i \right).$$

We found this variant to be quite effective on language modeling, reminiscent of the improvements observed by combining DeltaNet with Mamba2 [Gated DeltaNet; 82] in the linear attention case.

## 3 Efficient Training and Inference for PaTH Attention

Efficient kernels for attention [14, 12, 68] work by operating on subblocks of query and key matrices to avoid materialization of the full attention matrix in slower DRAM. Unlike in RoPE however, the cumulative products $\prod_s \mathbf{H}_s$ in PaTH are a function of the input and thus it is not clear whether PaTH-attention computations can similarly be decomposed into computations over subblocks. We now describe how the cumulative product of Householder[4] transformations can be efficiently computed using a compact representation of Householder products [27] and applied in a blockwise fashion [76, 47, 48, 81] to derive a FlashAttention-like algorithm that integrates blockwise Householder transformations with blockwise attention computations.

---

[3]However, this analogy is not quite as crisp in the Mamba2-FoX case. Mamba2 uses the recurrence $\mathbf{S}_t = f_t \mathbf{S}_{t-1} + \mathbf{v}_t \mathbf{k}_t^\top$, and unrolling this would give $\mathbf{o}_t = \sum_{j=1}^{t} \mathbf{v}_j \left( \prod_{s=j+1}^{t} f_s \right) \mathbf{k}_j^\top \mathbf{q}_t$. Applying softmax on this would give $\mathbf{o}_t = \frac{1}{Z_t} \sum_{j=1}^{t} \mathbf{v}_j \exp\left( \left( \prod_{s=j+1}^{t} f_s \right) \mathbf{k}_j^\top \mathbf{q}_t \right)$, which is different from FoX where the $\prod_{s=j+1}^{t} f_s$ term is outside the exponential function. In preliminary experiments we found this softmax version of Mamba2 to greatly underperform FoX.

[4]We hereon abuse terminology and use "Householder" to refer to our Householder-like transformations.

## 3.1 Background & Notation

We denote the block size along the sequence length dimension as $B$ and define subblocks using the notation $\mathbf{A}_{[i],[j]} := \mathbf{A}_{iB:(i+1)B, jB:(j+1)B} \in \mathbb{R}^{B \times B}$. This notation extends analogously to the other blocks $\mathbf{X}_{[i]} := \mathbf{X}_{iB:(i+1)B,:} \in \mathbb{R}^{B \times d}$ for $\mathbf{X} \in \{\mathbf{Q}, \mathbf{K}, \mathbf{V}, \mathbf{W}, \mathbf{O}\}$, where (for example) $\mathbf{W}_{[i]}$ is obtained from the vectors $\mathbf{w}_{iB}, \ldots, \mathbf{w}_{(i+1)B}$ in the Householder transformations.

**FlashAttention.** FlashAttention uses the online softmax trick [52, 61] to compute the output matrix $\mathbf{O}$ block by block. For each query block $i$ it sequentially process the key/value blocks $j$ from 0 to $i$, computing and accumulating the output as follows:

$$\mathbf{A}_{[i],[j]} \propto \begin{cases} \exp(\mathbf{Q}_{[i]}\mathbf{K}_{[j]}^\top), & \text{if } i < j \\ \exp(\text{lower}(\mathbf{Q}_{[i]}\mathbf{K}_{[i]}^\top)), & \text{if } i = j \end{cases} \in \mathbb{R}^{B \times B}, \qquad \mathbf{O}_{[i]} = \sum_{j=0}^{i} \mathbf{A}_{[i],[j]}\mathbf{V}_{[j]} \in \mathbb{R}^{B \times d}.$$

The attention submatrices $\mathbf{A}_{[i],[j]}$ are computed and processed entirely within SRAM, eliminating the need to write them to slower DRAM, which greatly reduces I/O costs and results in wallclock-speedups. Our algorithm also performs computations of the output block by block, but takes into account the additional contributions from the data-dependent Householder transformations.

**UT transform for products of Householder matrices.** A major challenge in computing PaTH attention lies in handling products of Householder matrices. We adopt the *UT transform* [27] to address this efficiently. For a sequence of $L$ transformations $\mathbf{H}_t = \mathbf{I} - \beta_t \mathbf{w}_t \mathbf{w}_t^\top$, their product can be compactly expressed as:

$$\mathbf{P} := \prod_{t=0}^{L-1} \mathbf{H}_t = \mathbf{I} - \mathbf{W}^\top \mathbf{T}^{-1}\mathbf{W} \qquad\qquad \in \mathbb{R}^{d \times d},$$

$$\text{where} \quad \mathbf{T}^{-1} := \left(\mathbf{I} + \text{strictLower}(\mathbf{D}\mathbf{W}\mathbf{W}^\top)\right)^{-1}\mathbf{D} \qquad\qquad \in \mathbb{R}^{L \times L}.$$

Here, $\mathbf{W} = [\mathbf{w}_0, \ldots, \mathbf{w}_{L-1}]^\top \in \mathbb{R}^{L \times d}$. $\mathbf{D} = \text{diag}([\beta_0, \ldots, \beta_{L-1}]) \in \mathbb{R}^{L \times L}$. We abuse notation for $\mathbf{T}^{-1}$ here for incorporating $\mathbf{D}$ to avoid notational clutter. The UT representation is efficient on modern hardware due to its use of triangular solves and matrix products [76], and is often preferred over alternatives such as the WY transform [5, 67].

## 3.2 Full Matrix Form of PaTH Attention

Recall that in PaTH attention, the attention score is given by $\mathbf{A}_{ij} \propto \exp\left(\mathbf{k}_j^\top \left(\prod_{t=j+1}^{i} \mathbf{H}_t\right) \mathbf{q}_i\right)$, which involves a cumulative product over arbitrary intervals $[j+1, i]$. A naïve implementation would require recomputing the UT transform for each such interval, which is computationally intractable. However, we show that it is possible to *reuse* the global matrix inverse $\mathbf{T}^{-1}$ and apply simple masking to efficiently extract the product over any subinterval.

To represent the product over an interval $\prod_{t=s_0}^{e_0} \mathbf{H}_t$ (with start index $s_0$ and end index $e_0$), we use the *masked UT transform*:

$$\prod_{t=s_0}^{e_0} \mathbf{H}_t = \mathbf{I} - (\mathbf{W} \odot \mathbf{M}_{s_0}^L)^\top \mathbf{T}^{-1}(\mathbf{W} \odot \mathbf{M}_{e_0}^R),$$

where $\odot$ denotes element-wise multiplication. The binary masks $\mathbf{M}_{s_0}^L, \mathbf{M}_{e_0}^R \in \mathbb{R}^{L \times d}$ are defined entrywise as:

$$(\mathbf{M}_{s_0}^L)_{k,c} = \begin{cases} 1 & \text{if } k \geq s_0, \\ 0 & \text{otherwise,} \end{cases} \quad (\mathbf{M}_{e_0}^R)_{k,c} = \begin{cases} 1 & \text{if } k \leq e_0, \\ 0 & \text{otherwise.} \end{cases}$$

Then, we have:

$$\widetilde{\mathbf{A}}_{ij} = \mathbf{k}_j^\top \left(\prod_{t=j+1}^{i} \mathbf{H}_t\right) \mathbf{q}_i = \mathbf{k}_j^\top \mathbf{q}_i - \mathbf{k}_j^\top (\mathbf{W} \odot \mathbf{M}_{j+1}^L)^\top \mathbf{T}^{-1}(\mathbf{W} \odot \mathbf{M}_i^R)\mathbf{q}_i$$

and equivalently, in matrix form:

$$\boxed{\widetilde{\mathbf{A}} = \text{lower}(\mathbf{Q}\mathbf{K}^\top) - \text{lower}(\mathbf{Q}\mathbf{W}^\top)\,\mathbf{T}^{-1}\,\text{strictLower}(\mathbf{W}\mathbf{K}^\top)}$$

This decomposition enables efficient pairwise attention computation using shared UT structure and interval-specific masking. However, computing the global inverse $\mathbf{T}^{-1}$ incurs a prohibitive $\mathcal{O}(L^3)$ time complexity with respect to sequence length $L$. In the following section, we introduce a blockwise algorithm that obtain the same result using only *local* inversions, thereby reducing the overall complexity to match that of standard attention mechanisms.

## 3.3 Efficient Training

To enable hardware-efficient (blockwise) training, cumulative Householder transformations must be pre-applied to the left and right boundaries of each block; otherwise, the token-specific nature of these transformations would render blockwise computation infeasible. To this end, we define boundary-adjusted query and key matrices as follows:

$$(\overleftarrow{\mathbf{Q}}_{[i]})_t = \left( \prod_{m=iB+1}^{iB+t} \mathbf{H}_m \right) \mathbf{q}_{iB+t} = \mathbf{q}_{iB+t} - \mathbf{W}_{[i]}^\top \mathbf{T}_{[i]}^{-1} (\mathbf{W}_{[i]} \odot \mathbf{M}_t^R) \mathbf{q}_{iB+t} \qquad \in \mathbb{R}^d,$$

$$(\overrightarrow{\mathbf{K}}_{[i]})_s = \left( \prod_{m=iB+s+1}^{(i+1)B} \mathbf{H}_m \right)^\top \mathbf{k}_{iB+s} = \mathbf{k}_{iB+s} - (\mathbf{T}_{[i]}^{-1} \mathbf{W}_{[i]})^\top (\mathbf{W}_{[i]} \odot \mathbf{M}_s^L) \mathbf{k}_{iB+s} \qquad \in \mathbb{R}^d,$$

a following the derivation in §3.2. In matrix form, these can be expressed as:

$$\overleftarrow{\mathbf{Q}}_{[i]} = \mathbf{Q}_{[i]} - \boxed{\text{lower}(\mathbf{Q}_{[i]} \mathbf{W}_{[i]}^\top)} \; \boxed{\mathbf{T}_{[i]}^{-1} \mathbf{W}_{[i]}} \qquad \in \mathbb{R}^{B \times d},$$

$$\overrightarrow{\mathbf{K}}_{[i]} = \mathbf{K}_{[i]} - \left( \boxed{\mathbf{T}_{[i]}^{-1} \text{strictLower}(\mathbf{W}_{[i]} \mathbf{K}_{[i]}^\top)} \right)^\top \mathbf{W}_{[i]} \qquad \in \mathbb{R}^{B \times d}.$$

With these quantities, we express the attention block computation as:

$$\mathbf{A}_{[i],[j]} \propto \begin{cases} \exp\left( \overleftarrow{\mathbf{Q}}_{[i]} \left( \prod_{m=j+1}^{i-1} \mathbf{P}_{[m]} \right)^\top \overrightarrow{\mathbf{K}}_{[j]}^\top \right), & \text{if } i > j, \\ \exp\left( \mathbf{Q}_{[i]} \mathbf{K}_{[i]}^\top - \boxed{\text{lower}(\mathbf{Q}_{[i]} \mathbf{W}_{[i]}^\top)} \; \boxed{\mathbf{T}_{[i]}^{-1} \text{strictLower}(\mathbf{W}_{[i]} \mathbf{K}_{[i]}^\top)} \right), & \text{if } i = j, \end{cases} \in \mathbb{R}^{B \times B},$$

where $\mathbf{P}_{[i]} := \prod_{j=1}^{B} \mathbf{H}_{iB+j} = \mathbf{W}_{[i]}^\top \boxed{\mathbf{T}_{[i]}^{-1} \mathbf{W}_{[i]}} \in \mathbb{R}^{d \times d}$. Due to associativity, the cross-block term can be computed *incrementally*: $\overleftarrow{\mathbf{Q}}_{[i]} \left( \prod_{m=j+1}^{i-1} \mathbf{P}_{[m]} \right)^\top \overrightarrow{\mathbf{K}}_{[j]} = (((\overleftarrow{\mathbf{Q}}_{[i]} \mathbf{P}_{[i-1]}^\top) \cdots) \mathbf{P}_{[j+1]}^\top) \overrightarrow{\mathbf{K}}_{[j]}.$

We adapt the FlashAttention-style block processing framework to perform a right-to-left scan over key/value blocks, enabling this product accumulation in a streaming manner. Concretely the modified blockwise workflow for processing query block $i$ is as follows:[5]

- Load $\overleftarrow{\mathbf{Q}}_{[i]}$ into SRAM.
- For key/value blocks $j = i - 1, \ldots, 0$ (right-to-left scan):
  - Load $\overrightarrow{\mathbf{K}}_{[j]}$, $\mathbf{V}_{[j]}$, and $\mathbf{P}_{[j]}$ from HBM into SRAM.
  - Compute logits: $\widetilde{\mathbf{A}}_{[i],[j]} = \overleftarrow{\mathbf{Q}}_{[i]} \overrightarrow{\mathbf{K}}_{[j]}^\top$.
  - Update online softmax statistics and accumulate output as in FlashAttention.
  - Update query: $\overleftarrow{\mathbf{Q}}_{[i]} \leftarrow \overleftarrow{\mathbf{Q}}_{[i]} \mathbf{P}_{[j]}^\top$.
- Normalize and store the output to HBM as in FlashAttention.

This design preserves the I/O efficiency of FlashAttention while incorporating PaTH's dynamic positional encoding via streaming cumulative products.

**Complexity analyses.** For each head, the attention computation between a pair of query and key blocks takes $\mathcal{O}(B^2 d + B d^2)$ time-$\mathcal{O}(B^2 d)$ for computing attention scores and $\mathcal{O}(B d^2)$ for

---

[5]Different query blocks can be executed in parallel, following a context-parallel strategy similar to that of FlashAttention-2 [12].

applying the transition on queries. Since there are $(L/B)^2$ such block pairs, the total attention cost is $\mathcal{O}(L^2 d + Ld^2/B)$. For preprocessing, computing the local Householder-based transformation for each query/key block involves an inversion step with cost $\mathcal{O}(B^3 + B^2 d)$. With $L/B$ such blocks, the total preprocessing cost is $\mathcal{O}(LB^2 + LBd)$. When $B \approx d$ (which is often the case), the overall complexity is comparable to standard attention, with quadratic scaling in sequence length.

**Speed Comparison.** We implement the PaTH attention kernel[6] in Triton [75] and benchmark its runtime on a single H100 GPU against FoX and standard RoPE attention under identical settings: batch size 32, 32 heads, head dimension 64, and varying sequence lengths. Results are shown in Figure 1. PaTH incurs a modest slowdown compared to RoPE, but outperforms FoX. Further speedups are expected from future kernel-level optimizations (e.g., via ThunderKittens [71]).

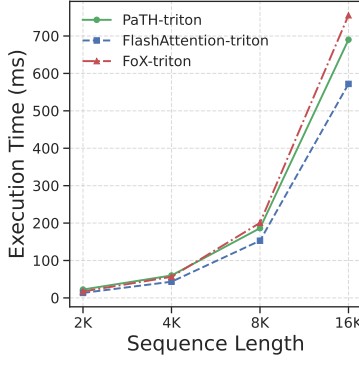

**Figure 1:** Speed comparison between attention variants.

### 3.4 Efficient Inference

We can efficiently update historical keys *in-place* using the current timestep's transition matrix:

$$\mathbf{k}_i^{(t)} \leftarrow (\mathbf{I} - \beta_t \mathbf{w}_t \mathbf{w}_t^\top)\mathbf{k}_i^{(t-1)} \quad \text{for all } i < t, \tag{1}$$

where $\mathbf{k}_i^{(i)} = \mathbf{k}_i$. This in-place update strategy eliminates the need to store a separate cache for $\{\mathbf{w}_i\}_{i \le t}$ or recompute the somewhat expensive cumulative Householder transformations. Then, the decoding stage becomes equivalent to standard softmax attention decoding, enabling compatibility with existing inference kernels such as FlashDecoding [15] and PagedAttention [34]. This approach maintains inference efficiency while preserving PaTH's dynamic positional encoding capabilities. Similarly, PaTH-FoX can be reduced to FoX decoding and thus compatible with the acceleration techniques of FoX (e.g., adaptive pruning [40]).

Before decoding, the initial key representations $\mathbf{k}_i^{(i)}$ must be transformed to $\mathbf{k}_i^{(l)}$ to account for subsequent Householder transformations. This transformation could be computed blockwise as:

$$\mathbf{K}_{[t]}^{(l)} = \overrightarrow{\mathbf{K}_{[t]}}\mathbf{P}_{[t+1]} \cdots \mathbf{P}_{[\lceil l/B \rceil]}.$$

It is also possible to reuse the suffix cumulative product $\mathbf{P}_{[t+1]} \cdots \mathbf{P}_{[\lceil l/B \rceil]}$ across blocks to reduce the overall complexity to linear.

### 3.5 Discussion

**Compatibility with context-parallelism (CP) techniques.** To extend our FlashAttention2-style context-parallel strategy to distributed settings such as Ring Attention [43, 38], PaTH's cumulative Householder transformations must be aligned with the ring-based key/value (KV) passing mechanism. Each device first precomputes its locally transformed queries ($\overleftarrow{\mathbf{Q}}$) and keys ($\overrightarrow{\mathbf{K}}$) by applying its resident Householder transformations. This also yields the local Householder product matrix $\mathbf{P}^{(d)}$ and softmax statistics for its sequence chunk. During inter-device communication, each device transmits its transformed $\overrightarrow{\mathbf{K}}$ vectors (with $\mathbf{V}$) and the associated $\mathbf{P}^{(d)}$ to the next device in the ring.

Upon receiving a $(\overrightarrow{\mathbf{K}}, \mathbf{V}, \mathbf{P}^{(d)})$ tuple from an earlier segment, the query-holding device first computes attention outputs using its current $\overleftarrow{\mathbf{Q}}$ and the incoming (transformed) keys, accumulating both the output and the corresponding online softmax statistics like standard attention. It then updates its $\overleftarrow{\mathbf{Q}}$ *in-place* via $\overleftarrow{\mathbf{Q}} \leftarrow \overleftarrow{\mathbf{Q}}(\mathbf{P}^{(d)})^\top$, propagating the cumulative path transformation forward along the ring. This sequence—compute output with current state, then update query state via incoming $\mathbf{P}^{(d)}$—faithfully emulates PaTH's logical right-to-left scan, enabling correct path reconstruction across distributed segments.

**Iterative refinement of KV cache.** From equation 1, PaTH iteratively applies low-rank updates to the historical key cache, forming a cumulative product of identity-plus-low-rank terms in the attention logit computation. This dynamic modification of the key cache is conceptually intriguing; see Song et al. [70], Ewer et al. [18], Leviathan et al. [35] for related ideas. Future directions include

---

[6] https://github.com/fla-org/flash-linear-attention/tree/main/fla/ops/path_attn

(i) extending this update mechanism to refine value vectors and (ii) developing more expressive yet hardware-efficient KV cache refinement schemes beyond the low-rank formulation used in PaTH.

# 4  Experiments

We experiment with PaTH attention and compare it against various baselines: ordinary RoPE attention, Stick-Breaking Attention (SBA) [73], and Forgetting Transformer (FoX) [39].

## 4.1  Synthetic Tasks

**Flip-flop language modeling.**  We first experiment with *flip-flop language modeling* (FFLM) [41], a diagnostic synthetic task which has been found to be challenging for existing architectures. In this task, the vocabulary consists of $\Sigma = \{\mathtt{w,r,i,0,1}\}$. Given a sequence of write-bit, read-bit, ignore-bit actions, the model must produce the bit (0 or 1) after the most recent write-bit action. For example given the sequence "w 1 r 1 w 0 i 1 i 0 i 1 r", the model is expected to recall the most recently written bit, i.e., 0. Despite its simplicity, flip-flop language modeling is diagnostic of many real-world capabilities, such as modeling long-range dependencies, the ability to ignore distractors, and sequential reasoning. Liu et al. [41] find that RoPE-based transformers struggle on this task and provide theoretical insights into why RoPE-based attention mechanisms find it inherently difficult. In Theorem A.1 of the appendix we show that there exists a 2-layer PaTH-based transformer that can solve this task. Empirically, our experiments in Table 1 show that PaTH-based transformers can practically learn to almost perfectly solve this task with only a single layer and two attention heads, including out-of-distribution settings whose frequency of operations are different from than in training (sparse means 98% of the operations are ignore, while dense means only 10% are ignore).

| Method | ID | OOD | |
|---|---|---|---|
| | | **Sparse** | **Dense** |
| RoPE | 6.9% | 40.3% | 0.01% |
| SBA [73] | 9.6% | 38.9% | 0% |
| FoX [39] | 8.3% | 36.3% | 0% |
| PaTH | 0% | 0.0001% | 0% |

**Table 1:** FFLM error rate (%) on ID/OOD test sets. All models are 1-layer, 2-head, 64-dim.

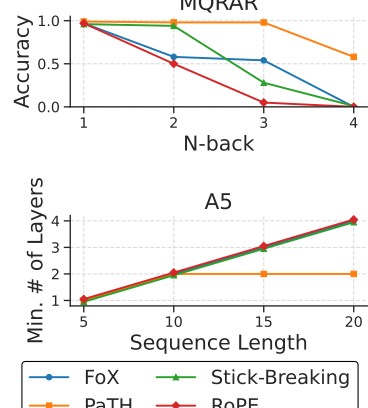

**Figure 2:** Results on MQRAR-N (top) and $A_5$ word problem (bottom).

**Word problems.**  We showed in §2.2 that PaTH can theoretically extend transformers beyond $\mathsf{TC}^0$. However, it is a different question as to whether PaTH transformers can empirically *learn* to solve $\mathsf{NC}^1$-complete problems based on actual data. To test this, we follow Merrill et al. [51] and use a word problem task based on the alternating group $A_5$, a subgroup of $S_5$ (on which the word problem is also $\mathsf{NC}^1$-complete). This task requires determining if a "word"—a sequence of group operations using fixed generators and their inverses—evaluates to the identity element. Successfully performing this symbolic task means the model must implicitly learn algebraic rules like permutation composition and cancellation. As a concrete example, consider generators $g_1 = (1\,2\,3)$, $g_2 = (1\,2\,4)$, and $g_3 = (1\,2\,5)$, with their respective inverses $g_1^{-1}, g_2^{-1}, g_3^{-1}$. Given the word $w = g_1 \cdot g_2 \cdot g_1^{-1} \cdot g_2^{-1}$, the model must determine if $w$ equals the identity permutation. In this instance, $w$ is not the identity, and the model needs to correctly track the sequence of permutations to arrive at this conclusion. Figure 2 (bottom) shows that PaTH can solve this task defined as achieving above 90% acciracy following Merrill et al. [51]) with fewer layers than baselines.

**Multi-query Repeated Associative Recall with *N*-back (MQRAR-*N*).**  We adapt the Multi-query Repeated Associative Recall (MQRAR) task from Tan et al. [73] (itself an enhancement of MQAR [1]) to MQRAR-*N*-back. This task tests a model's associative recall ability by requiring it to find the $N$-th last assignment for a given variable, drawing an analogy to the *N-back* task in experimental psychology [30]. Recalling the most recent assignment ($N = 1$) can often be accomplished by simpler, recency-focused mechanisms. However, retrieving the $N$-th last assignment ($N > 1$) more rigorously probes a model's capacity to track an ordered history of states for specific variables, especially when recent information must be ignored. An example sequence for $N = 2$ is:

| **Input** | A | 1 | B | 2 | C | 3 | D | 4 | A | 5 | B | 6 | A | 7 | C | 8 | A | 9 | B | 0 |
|---|---|---|---|---|---|---|---|---|---|---|---|---|---|---|---|---|---|---|---|---|
| **Output** | $\phi$ | $\phi$ | $\phi$ | $\phi$ | $\phi$ | $\phi$ | $\phi$ | $\phi$ | $\phi$ | $\phi$ | $\phi$ | $\phi$ | 1 | $\phi$ | $\phi$ | $\phi$ | 5 | $\phi$ | 2 | $\phi$ |

We compare Transformer models using RoPE, SBA, FoX, and PaTH on their ability to handle MQRAR-$N$-back with $N \in \{1, 2, 3, 4\}$. All models are 2-layer Transformers with a 256-dimensional hidden state, 2 attention heads. For the task we use 32 key-value pairs a sequence length of 768. Figure 2 shows the results, where we find that PaTH attention can successfully track variable values with $N$-back recall for $N < 4$, whereas recent baselines (SBA and FoX) still struggle.

## 4.2 Language Modeling

We pretrain language models with $\sim$760M parameters on the Fineweb-Edu corpus [54] for 50B tokens using the Mistral tokenizer and a sequence length of 4096. We then evaluate the pretrained models on the following benchmarks. See appendix B for full details and additional experiments.

| Model | Wiki. ppl $\downarrow$ | LMB. ppl $\downarrow$ | LMB. acc $\uparrow$ | PIQA acc $\uparrow$ | Hella. acc_n $\uparrow$ | Wino. acc $\uparrow$ | ARC-e acc $\uparrow$ | ARC-c acc_n $\uparrow$ | Avg. $\uparrow$ |
|---|---|---|---|---|---|---|---|---|---|
| RoPE | 19.01 | 19.77 | 40.4 | 70.2 | 50.3 | 54.9 | 67.2 | 33.3 | 52.7 |
| FoX | 18.33 | 18.28 | 41.7 | **70.8** | 50.9 | **57.1** | 65.7 | 32.6 | 53.1 |
| PaTH | 18.03 | 16.79 | 44.0 | 70.5 | 51.5 | 56.0 | **68.9** | **34.4** | **54.2** |
| PaTH-FoX | **17.35** | **16.23** | **44.1** | **70.8** | **52.2** | **57.1** | 67.3 | 33.9 | **54.2** |

**Table 2:** Results on perplexity and zero-shot commonsense reasoning tasks for 760M models trained on 50B tokens. Best results are highlighted in bold, while the second best results underlined.

**Standard LM benchmarks.** We evaluate on Wikitext perplexity and selected zero-shot common sense reasoning tasks, including of LAMBADA [LMB.; 53] (OpenAI version), PiQA [6], HellaSwag [Hella.; 83], WinoGrande [Wino.; 64], ARC-easy (ARC-e) and ARC-challenge (Arc-c) [10]. Table 2 shows the results. PaTH consistently outperforms RoPE across all tasks, and surpasses FoX on most. PaTH-FoX performs comparably with PaTH while achieving the lower perplexity.

**Length extrapolation.** Figure 3 presents results on three long-context corpora from different domains: PG-19 [62] (books), CodeParrot (code), and NarrativeQA [31](conversational English). Both PaTH-FoX and FoX generalize up to 64K tokens, with PaTH-FoX consistently achieving lower perplexity. The improvement is especially pronounced in the code domain, where state tracking—e.g., tracking variable values—is crucial. PaTH alone generalizes reasonably well, maintaining stable performance up to 32K tokens, after which perplexity gradually increases (in contrast to RoPE, which fails abruptly beyond 4K). These results underscore the benefit of data-dependent position encoding and the critical role of the forgetting mechanism in enabling robust generalization to longer contexts.

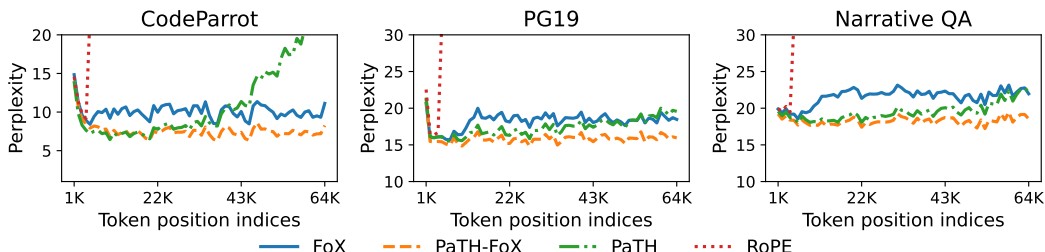

**Figure 3:** Length extrapolation results for 760M models trained on 50B tokens with 4096 context length.

**Long-context benchmarks.** Table 3 summarizes results on four challenging long-context benchmarks: RULER [23], BABILONG [33], PhoneBook [26], and LongBench-E [3]. For RULER, we report the zero-shot average accuracy across all 13 subtasks and also breakdowns by task categories and context length in Figure 4; for BABILONG, we follow standard practice and report the average few-shot accuracy over subproblems QA0–QA5 (see Figure 5 for breakdowns by task and context length); for LongBench-E, we report average scores across three length intervals—0–4K, 4–8K, and 8–16K—and provide detailed results in Table 7.

These benchmarks assess different aspects of long-context understanding. Accurate retrieval is critical and is tested by RULER's Single- and Multi- Needle-In-A-Haystack (NIAH) tasks, as well as by PhoneBook Lookup, an extreme case where every token in the context is a 'needle'. PaTH-FoX achieves the highest overall retrieval performance, excelling in the more difficult Multi-NIAH and PhoneBook settings.

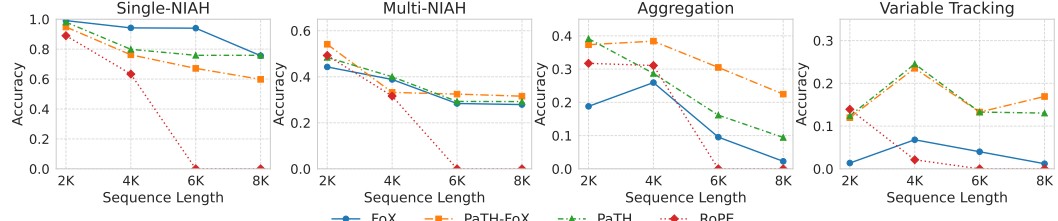

**Figure 4:** RULER results grouped by different task categories.

| Model | RULER | | | BABILONG | | | | PhoneBook | | | LongBench-E | | |
|---|---|---|---|---|---|---|---|---|---|---|---|---|---|
| | 4K | 8K | 16K | 0K | 4K | 8K | 16K | 2K | 4K | 8K | 4K | 8K | 16K |
| RoPE | 35.7 | 1.3 | 0.0 | 33.0 | 13.8 | 0.0 | 0.0 | 32.3 | 15.6 | 0.0 | 18.7 | 3.7 | 2.0 |
| FoX | 41.6 | 29.5 | 4.9 | 23.8 | 20.2 | 8.2 | 4.4 | 62.5 | 38.5 | 17.7 | 23.4 | 16.9 | 11.7 |
| PaTH | **44.6** | **34.8** | 18.7 | **33.8** | 24.6 | 16.8 | **11.6** | 55.2 | 20.8 | 0.0 | **27.2** | **22.5** | 14.4 |
| PaTH-FoX | 42.3 | 34.0 | **22.6** | 28.6 | **25.6** | **19.2** | 10.0 | **89.6** | **93.8** | **66.6** | 23.4 | 21.8 | **16.1** |

**Table 3:** Summary of average scores on long-context tasks for 760M models with training length 4096.

Beyond retrieval, RULER also probes state track-ing through its Variable Tracking (VT) task.[7] PaTH and PaTH-FoX achieve substantial gains here, consis-tent with their advantages on synthetic state-tracking tasks. BABILONG further tests such capabilities in a narrative setting, embedding bAbI-style logic queries within long PG-19 passages—thus requiring both en-tity tracking and multi-hop reasoning over extended text. On these tasks as well, PaTH and PaTH-FoX clearly outperform FoX and RoPE.

| Model | GSM8K | HumanEval | MBPP+ |
|---|---|---|---|
| RoPE | 19.9 | 23.1 | 47.1 |
| FoX | 15.5 | 21.3 | 48.2 |
| PaTH | **20.1** | **25.6** | **51.3** |
| Base | 8.6 | 16.4 | 38.6 |

**Table 4:** Results on math and coding benchmarks after conversion. *Base* denotes the teacher model performance before continued pretraining.

### 4.3 Converting RoPE into PaTH

Training LLMs from scratch is highly resource-intensive. We hence explore *con-verting* pretrained RoPE-based LLMs into PaTH-based LLMs, in particular targeting improvements in math/coding domains.

Following Goldstein et al. [20], we use a two-stage distillation process first minimizes the Mean Squared Error (MSE) between the attention-layer outputs of the RoPE teacher and the PaTH student, followed by fine-tuning using KL divergence on the outputs. The first and second stages use 100M and 3B tokens, respectively, from the DCLM corpus

| Task | Teacher (RoPE) | Student (PaTH) |
|---|---|---|
| MMLU | **74.21** | 73.28 |
| HellaSwag | **85.20** | 84.83 |
| Winogrande | **71.51** | 68.90 |
| GPQA Diamond | 33.33 | **34.34** |
| TheoremQA | 18.12 | **21.88** |
| GSM-8K | 80.29 | **80.67** |
| MATH | **69.10** | 65.38 |
| HumanEval | **82.32** | 77.44 |
| MBPP | 74.71 | **75.10** |
| RULER (4K) | **94.37** | 93.24 |

**Table 5:** `Qwen2.5-7B-Instruct` distillation results (with-out continued pretraining on math/code data).

[37]. After distillation, we perform continued pretraining using a balanced mixture (1:1:1) of DCLM (text), Python-Edu (code), and MegaMathWeb (math) corpora [87] of 21B tokens. Since it may be difficult to observe sizeable improvements over existing (often overtrained) state-of-the-art models that have already been exposed to extensive math/coding data, we work with the `SmolLM2-1.7B` checkpoint[8] taken immediately before the WSD decay stage [24], i.e., prior to exposure to high-quality math and code data. As shown in Table 4, PaTH consistently outperforms both RoPE and FoX. We speculate that PaTH's expressivity and state-tracking capabilities contribute to its advantages in handling math and coding tasks.

While the above results are promising, we find mixed results when distilling from models that have already been extensively (over)trained. Table 5 shows the performance when distilling `Qwen2.5-7B-Instruct` [60] without the continued pretraining stage: PaTH student can improve the teacher's performance across some benchmarks, but there is degradation across others. These

---

[7]E.g., given "`VAR X1 = 12345, VAR X2 = 3212, ..., VAR X10 = X1, ...`" the query might ask "`Find all variables assigned the value 12345`", with the correct answer being "`X1, X10`".

[8]https://huggingface.co/HuggingFaceTB/SmolLM2-nanotron-ckpt/tree/main/1700M/pre-decay

distillation experiments suggest that it may be important to start the conversion process before the original model (potentially) ossifies and becomes difficult to convert; better conversion recipes remain an avenue for future work.

## 5    Related Work

**Data-dependent positional encoding.**    RoPE [72] has been the *de facto* position encoding scheme in large language models. However, RoPE's static nature makes it unsuitable for dynamically adapting to long sequences, motivating works on RoPE length extension [55, 8, 44, *inter alia*]. Yet, these methods remain within the RoPE framework and can only mitigate rather solve its limitations. An alternative line of work focuses on *data-dependent* position encoding. While promising, these approaches operate solely at the attention logit level, modifying the $\mathbf{QK}^\top$ scores through post hoc transformations [85, 39, 86, 21, 35, 73, 11]. However, the dot-product structure is fundamentally limited in its ability to represent more intricate dependencies [19, 32], motivating work on *algebraic position encodings* [32], where relative positions are encoded via cumulative matrix products. While conceptually similar to our approach, APE focuses exclusively on data-*independent* orthogonal (and thus invertible) matrices that are simultaneously diagonalizable [59], and thus inherently limited in expressivity [9, 51, 74]. In contrast, our proposed PaTH method addresses this limitation by using *data-dependent* cumulative Householder-like products, which are non-invertible, non-commutative, and not simultaneously diagonalizable, leading to more expressive transformations of the unnormalized attention logits. Moreover, PaTH is compatible with other attention variants, such as FoX, providing a principled and extensible framework for positional encoding.

**Improving state tracking in language models.**    Transformer-based language models often struggle with state and entity tracking [29, 57, 51]. This is potentially due to the standard transformer architecture's finding it difficult to reliably emulate finite-state automata [41, 42, 88, 4]. To shed light on the theoretical reasons transformers struggle with word problems (tasks requiring careful state tracking), recent studies have analyzed their learning dynamics [36] and conducted mechanistic investigations [84]. Researchers have also proposed alternative attention mechanisms to enhance self-attention's expressivity. These aim to capture richer pairwise dependencies than standard dot-product attention, often by incorporating lightweight recurrence—such as right-to-left cumulative sums—into the attention logits [21, 35, 73]. Fagnou et al. [19] propose a matrix-inversion-based attention mechanism for capturing path-level dependencies, which is conceptually similar to our approach. While these methods show empirical improvements in state or entity tracking tasks, they are largely heuristic. In this work, we draw inspiration from theoretical studies on parallelizing RNNs while preserving their state tracking capabilities [51, 22, 69, 56]. From these, we design a new softmax-based attention mechanism that is performant and efficient.

## 6    Limitation

While PaTH improves expressivity, it has several practical caveats. Training stability can be sensitive to numerical precision. In particular, the cumulative product of Householder transformations may become unstable under BF16, requiring clipping of the scaling factor $\beta$ to prevent it from reaching 2, as BF16 rounding can otherwise produce eigenvalues larger than 1 and cause divergence. In addition, the speed comparisons in this work are restricted to head dimension 64. Larger head dimensions increase the computational and memory overhead of PaTH. Finally, PaTH does not directly model rotations, as a single reflection matrix does not subsume rotational transformations. This may limit certain geometric inductive biases present in RoPE, which arise from its rotation-based structure, such as its structured dependence on relative position. Extending PaTH with compositions of reflections to approximate rotations, similar in spirit to DeltaProduct [69], is an interesting direction for future work.

## 7    Conclusion

This work introduces PaTH, a new data dependent multiplicative position encoding scheme that provably enhances the expressive power of Transformers. We develop a FlashAttention style blockwise algorithm to enable efficient parallel training. Experiments show that PaTH consistently outperforms RoPE across multiple benchmarks, with particularly strong gains on state tracking tasks and length extrapolation.

## Acknowledgements

This study was supported in part by the AI2050 program at Schmidt Sciences (Grant G-25-67980), MIT-IBM Watson AI Lab, and the CSAIL Felicis Research Program. We also thank Zhixuan Lin for helpful discussions.

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

# A  Representation Power of Transformers with PaTH Attention

We state two theorem which illustrate the representation power of transformers equipped with PaTH attention.

The first theorem shows that a PaTH attention layer can solve the problem of tracking iterative swaps on 5 elements, which is an $\mathsf{NC}^1$-complete under $\mathsf{AC}^0$ reductions. This theorem and its proof is an adaptation of Theorem 2 of Peng et al. [56].

**Theorem 2.1.** *A one-layer PaTH transformer with two attention heads and* $\log n$ *precision can solve an* $\mathsf{NC}^1$*-complete problem under* $\mathsf{AC}^0$*-reductions.*

*Proof.* As in Lemma 2 of Peng et al. [56], consider the task of deciding whether $n$ iterative swappings of 5 elements encodes the identity permutation. This task consists of an input sequence $c = c_0 c_1 \ldots c_n$ of length $n + 1$,

$$\# \ [a_1 \leftrightarrow b_1] \ [a_2 \leftrightarrow b_2] \ \ldots \ [a_n \leftrightarrow b_n]$$

where $c_0 = \#$ is the start token and $c_1 = [a_1 \leftrightarrow b_1], \ldots, c_n = [a_n \leftrightarrow b_n]$ are "tokens" which indicates that position $a_n$ is swapped with position $b_n$ at time $n$. (Hence there are 20 such possible swap tokens of the form $[x \leftrightarrow y]$ for all pairwise $x, y \in \{1, \ldots, 5\}$ such that $x \neq y$.) Given this sequence, we show that there is a one-layer PaTH transformer with two attention heads that outputs a 1 if the sequence encodes the identity permutation, and $-1$ otherwise. As noted by previous works [51, 56], this suffices since there is an $\mathsf{AC}^0$-reduction from a well-known $\mathsf{NC}^1$-complete problem (i.e., iterated multiplication of $S_5$) to this task.

We first embed the $\#$ and all 20 $[x \leftrightarrow y]$ tokens to distinct one-hot vectors. Given a token $u \in \Sigma$ and its associated one-hot vector $\mathbf{u}$, we choose the key/query/value/PaTH projection matrices (i.e., $\mathbf{W}_k, \mathbf{W}_q, \mathbf{W}_v, \mathbf{W}_w \in \mathbb{R}^{6 \times 21}$) matrices for the first attention head such that

$$\mathbf{W}_k \mathbf{u} = \mathbf{k}[u] = \mathbf{1}\{u = \#\}(\mathbf{e}_1 + 2\mathbf{e}_2 + 3\mathbf{e}_3 + 4\mathbf{e}_4 + 5\mathbf{e}_5 - \mathbf{e}_6),$$
$$\mathbf{W}_q \mathbf{u} = \mathbf{q}[u] = n(\mathbf{e}_1 + 2\mathbf{e}_2 + 3\mathbf{e}_3 + 4\mathbf{e}_4 + 5\mathbf{e}_5 + 54.5\mathbf{e}_6),$$
$$\mathbf{W}_w \mathbf{u} = \mathbf{w}[u] = (\mathbf{e}_x - \mathbf{e}_y)/\sqrt{2} \text{ for } v = [x \leftrightarrow y], \text{ and } \mathbf{0} \text{ if } v = \#,$$
$$\mathbf{W}_v \mathbf{u} = \mathbf{v}[u] = \mathbf{1}\{u = \#\}\mathbf{e}_1$$
$$\beta = 2.$$

(Hence, the query vectors and $\beta$ are input-independent.) In this case, as in Lemma 1 of [56] the one-step PaTH transformation is a true Householder transformation with

$$\mathbf{H}[u] = \mathbf{I} - 2\mathbf{w}[u]\mathbf{w}[u]^\top \in \mathbb{R}^{6 \times 6}$$

and effectively swaps $x$ with $y$. Now suppose the initial list is $[1, 2, 3, 4, 5]$, and let $\pi(i)$ be the $i$-th element of the final permuted list after the $n$ swaps. We then have

$$(\mathbf{k}[c_0]^\top \prod_{s=1}^{n} \mathbf{H}_s) = \left(\left(\sum_{i=1}^{5} i\mathbf{e}_{\pi(i)}\right) - \mathbf{e}_6\right)^\top,$$

and the attention logit from $n$ to 0 is given by

$$s_0 = \mathbf{k}[c_0]^\top \prod_{s=1}^{n} \mathbf{H}_s \mathbf{q}[c_n] = n\left(\sum_{i=1}^{5} i\pi(i) - 54.5\right).$$

By the rearrangement inequality, we further have

$$\sum_{i=1}^{5} i\pi(i) \leq \sum_{i=1}^{5} i^2 = 55,$$

with equality holding if and only if $i = \pi(i)$ for all $i$. Therefore $s_0 > 0.5n$ if the final list is the same as the initial list (i.e., identity permutation), and $s_0 < -0.5n$ otherwise. Because $\mathbf{k}[u] = \mathbf{0}$ for all $u \neq \#$, we further have that the attention logits $s_l$ for all $l > 0$ is 0. The attention weight for the first position is then given by $a_0 = \frac{\exp(s_0)}{\exp(s_0) + n}$, which is greater than $\frac{1}{n+1}$ if $s_0 > 0$ (i.e., permutation is

identity) and less than $\frac{1}{n+1}$ otherwise. Since the value vector is $\mathbf{e}_1$ for $c_0$ and $\mathbf{0}$ otherwise, the output of this attention head is given by

$$\sum_{l=0}^{n} a_l \mathbf{v}[c_l] = \frac{\exp(s_0)}{\exp(s_0) + n} \mathbf{e}_1.$$

The second attention head is data-independent and uses $\mathbf{W}_k = \mathbf{W}_q = \mathbf{W}_w = \mathbf{0}$, and the same value matrix $\mathbf{W}_v$ as above. This results in the output of this second attention head always being $\frac{1}{n+1}\mathbf{e}_1$ regardless of the input. Concatenating the output from these two heads gives the vector

$$\left[\frac{\exp(s_0)}{\exp(s_0) + n}, 0, 0, 0, 0, 0, \frac{1}{n+1}, 0, 0, 0, 0, 0\right],$$

i.e., 12 dimension vector with the first dimension as $\frac{\exp(s_0)}{\exp(s_0)+n}$ and the 7th dimension as $\frac{1}{n+1}$. We can now have an output projection layer with matrix $\mathbf{W}_o$ that subtracts the 7th dimension from the 1st dimension (i.e., $[1, 0, 0, 0, 0, 0, -1, 0, 0, 0, 0, 0]$ in the first row). The first dimension of this output vector will be positive if the permutation is identity, and negative otherwise. We can then use the FFN layer with a $\text{sign}(\cdot)$ nonlinearty (or a steep tanh function) to clamp this output to $\{-1, +1\}$.

We do not explicitly need the $\log n$ precision assumption here but the construction here can be represented in $\log n$ precision while preserving the same functionality. We include this assumption to ensure that we are using same or weaker precision assumption with previous works on the circuit complexity of transformers (Merrill and Sabharwal [50], Chen et al. [7] and refs. therein). We can make the proof simpler in the above if we incorporate a $O(\log n)$ assumption since in this case the output of softmax is 1 when the final list is the same as the original list and is 0 otherwise (i.e., there is no need for the second attention head). □

**Theorem A.1.** *For any $n$, there is a two-layer PaTH transformer with $O(\log n)$ precision can solve the flip-flop language modeling (FFLM) task with accuracy greater than $1 - 1/n^{100}$ for all inputs up to length $n$.*

*Proof.* Recall that in FFLM, there are five types of input `w`, `i`, `r`, `0`, `1`. We will now present a construction of the two-layer transformer with PaTH attention.

The token embeddings are given by

$$\begin{aligned}
\text{emb}(\mathtt{w}) &= \mathbf{e}_1 + \mathbf{e}_6 \\
\text{emb}(\mathtt{r}) &= \mathbf{e}_2 + \mathbf{e}_6 \\
\text{emb}(\mathtt{i}) &= \mathbf{e}_3 + \mathbf{e}_6 \\
\text{emb}(\mathtt{0}) &= \mathbf{e}_4 + \mathbf{e}_6 \\
\text{emb}(\mathtt{1}) &= \mathbf{e}_5 + \mathbf{e}_6
\end{aligned}$$

where $\mathbf{e}_i$ is the one-hot $i$-th basis vector.

The first attention layer will implement a one-hot attention from the bit tokens `0` and `1` to their corresponding instruction tokens. To achieve this, we will have the matrices $\mathbf{W}_k, \mathbf{W}_q, \mathbf{W}_w, \mathbf{W}_v$ such that:

$$\begin{aligned}
\mathbf{W}_k \mathbf{h} &= (h_1 + h_2 + h_3)\mathbf{e}_1, \\
\mathbf{W}_q \mathbf{h} &= n h_6 \mathbf{e}_1, \\
\mathbf{W}_w \mathbf{h} &= (h_1 + h_2 + h_3)\mathbf{e}_1 + (h_4 + h_5)\mathbf{e}_2, \\
\mathbf{W}_v \mathbf{h} &= h_1 \mathbf{e}_7 + h_2 \mathbf{e}_8 + h_3 \mathbf{e}_9, \\
\beta &= 1.
\end{aligned}$$

Then the transition matrix is given by

$$\mathbf{H} = \begin{cases} \mathbf{I} - \mathbf{e}_1 \mathbf{e}_1^\top, & \text{if input is } \{\mathtt{w}, \mathtt{r}, \mathtt{i}\} \\ \mathbf{I} - \mathbf{e}_2 \mathbf{e}_2^\top, & \text{if input is } \{\mathtt{0}, \mathtt{1}\}, \end{cases}$$

i.e., the transition matrix projects the first dimension to $0$ for the instruction tokens $\{w, r, i\}$ and projects the second dimension to $0$ the bit tokens $\{0, 1\}$. Similarly, the key vector $\mathbf{k}_i$ is $\mathbf{e}_1$ if the $i$-th token is an instruction token, and $\mathbf{0}$ otherwise. Therefore when the $i$-th token is $0$ or $1$,

$$\mathbf{k}_j^\top \prod_{s=j+1}^{i} \mathbf{H}_s \mathbf{q}_i \neq 0$$

if and only if $j = i - 1$, and in this case it equals to $n$. Because we are considering an $O(\log n)$ precision transformer, the attention score after softmax becomes 1-hot for every bit token. After this attention layer, the 7-th to 9-th dimension of the bit tokens now encode the type of instruction of the previous token.

The first FFN layer will map the 1 to 9 dimensions of $0$ and $1$ tokens to be a one-hot embedding for each value and corresponding instruction type,

$$\text{FFN}(\mathbf{h})_i = 0, i \notin \{10, 11, 12\},$$
$$\text{FFN}(\mathbf{h})_{10} = \mathbf{1}\{h_4 = 1, h_7 = 1\},$$
$$\text{FFN}(\mathbf{h})_{11} = \mathbf{1}\{h_5 = 1, h_7 = 1\},$$
$$\text{FFN}(\mathbf{h})_{12} = 1, \text{ otherwise.}$$

With $\mathbf{1}\{\cdot\}$ being the indicator function. Specifically, the 10-th dimension will be 1 for every $0$ following a $w$ and the 11-th dimension will be 1 for every $1$ following a $w$.

The second attention layer will operate on the 10-th and 11-th dimensions of the input embedding and implement the following:

$$\mathbf{W}_k \mathbf{h} = (h_{10} + h_{11})\mathbf{e}_1$$
$$\mathbf{W}_q \mathbf{h} = n h_6 \mathbf{e}_1$$
$$\mathbf{W}_w \mathbf{h} = (h_{10} + h_{11})\mathbf{e}_1 + h_{12}\mathbf{e}_2$$
$$\mathbf{W}_v \mathbf{h} = h_8 \mathbf{e}_{13} + h_9 \mathbf{e}_{14}$$
$$\beta(\mathbf{h}) = \mathbf{1}\{h_8 + h_9 > 0\}$$

Here we assume that we can use a step function for $\beta$ (or alternatively, we can use a steep-enough logistic function for it to be effectively a step function under the precision considered). This shows that for every token that is not a $0$ or $1$ that follows $w$, the transition matrix is identity; for $0$ or $1$ that follows $w$, the transition matrix is a matrix that projects the first dimension to $0$. Then for any $i \geq 2$,

$$\mathbf{k}_j^\top \prod_{s=j+1}^{i} \mathbf{H}_s \mathbf{q}_i \neq 0$$

if and only if $j$ is the largest token that is a $0$ or $1$ that follows $w$ with $j \leq i$. This $j$ is guaranteed to exist because in FFLM, the first token is always $w$. In this case, this term equals $n$. Using the same argument as the first layer, the attention becomes one-hot and the output of attention encode the value of last $0$ or $1$ token following a $w$. By the definition of flip-flop, this is the current state.

The second FFN layer will operate on the 13-th and 14-th dimensions of the input,

$$\text{FFN}(\mathbf{h})_i = 0, i \notin \{15, 16\},$$
$$\text{FFN}(\mathbf{h})_{15} = \mathbf{1}(h_2 = 1, h_6 = 1),$$
$$\text{FFN}(\mathbf{h})_{16} = \mathbf{1}(h_2 = 1, h_6 = 1).$$

Specifically, the 15-th and 16-th dimension of the output will encode the state value for each $r$ token. After this layer, dimensions 1, 3, 4, 5, 15, and 16 of the embedding becomes one-hot, each corresponding to a different output distribution in FFLM.

Finally, the LM head will map dimensions 1, 3, 4, 5, 15, and 16 to their corresponding next-token probability before softmax. Concretely,

$$\mathbf{W}_{LM}\mathbf{h} = (T\mathbf{e}_4 + T\mathbf{e}_5)(h_1 + h_3)$$
$$+ (T\mathbf{e}_1 + T\mathbf{e}_2 + T\mathbf{e}_3)(h_4 + h_5)$$
$$+ n\mathbf{e}_4 h_{15} + n\mathbf{e}_5 h_{16}.$$

Here $T \approx \log n$ is an appropriate number such that softmax over $T\mathbf{e}_4 + T\mathbf{e}_5$ and $T\mathbf{e}_1 + T\mathbf{e}_2 + T\mathbf{e}_3$ yields a uniform distribution with error smaller than $1/n^{101}$. $\qquad \square$

| Task | Example | Evaluation Focus |
|------|---------|------------------|
| Task 1: Single Supporting Fact | Mary went to the bathroom.
John moved to the hallway.
Mary travelled to the office.
**Q: Where is Mary? A: office** | Identify a single explicit fact from context. |
| Task 2: Two Supporting Facts | John is in the playground.
John picked up the football.
Bob went to the kitchen.
**Q: Where is the football? A: playground** | Combine two clues to infer an object's location. |
| Task 3: Three Supporting Facts | John picked up the apple.
John went to the office.
John went to the kitchen.
John dropped the apple.
**Q: Where was the apple before the kitchen? A: office** | Track object movement and temporal order. |
| Task 4: Two Argument Relations | Office is north of bedroom.
Bedroom is north of bathroom.
Kitchen is west of garden.
**Q1: What is north of bedroom? A: office**
**Q2: What is bedroom north of? A: bathroom** | Reason over spatial relationships. |
| Task 5: Three Argument Relations | Mary gave the cake to Fred.
Fred gave the cake to Bill.
Jeff was given the milk by Bill.
**Q1: Who gave the cake to Fred? A: Mary**
**Q2: Who did Fred give the cake to? A: Bill** | Transitive reasoning over possession chains. |

**Table 6:** Descriptions and examples of the first five bAbI tasks. Each task highlights a specific reasoning skill required for successful question answering.

# B   Experimental Setup & Additional Results

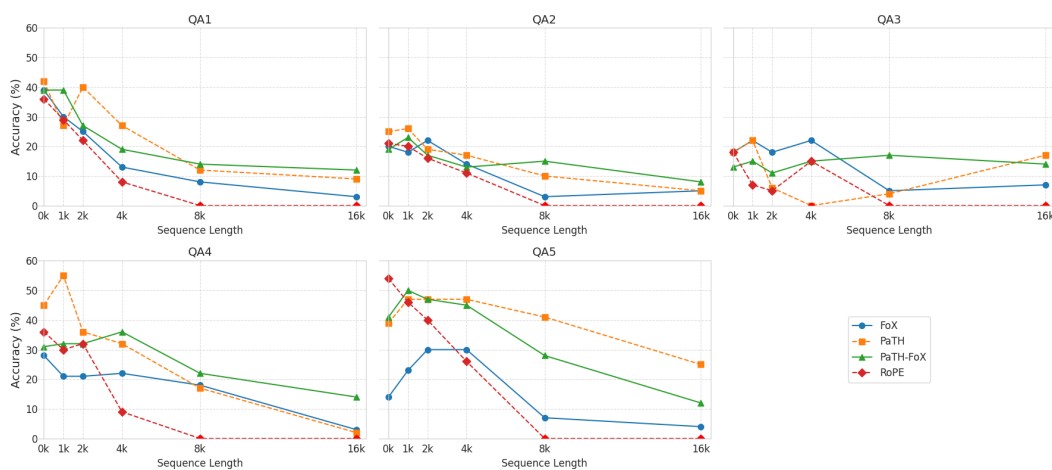

**Figure 5:** BABILong performance breakdowns. QA1: Single supporting fact. QA2: Two supporting facts. QA3: Three supporting facts. QA4: Two arg relations. QA5: Three arg relations.

**Hyperparameter settings.**   All models are trained with AdamW [46], using a cosine learning rate schedule with a 1B-token warmup. The peak learning rate is 1e-3, with both initial and final rates set to 3e-5. We apply a weight decay of 0.01 and gradient clipping of 1.0. The batch size is 2M tokens. Parameters are initialized with a standard deviation of 0.02. Each 760M model is trained on 8 H100 GPUs for 2-3 days. For synthetic tasks, we use A100 GPUs, completing training within several hours.

**BABILong**   Figure 5 presents the performance breakdown across sub-tasks and sequence lengths. Task descriptions are provided in Table 6.

**LongBench-E**  Detailed results are presented in Table 7.

| Category | Dataset | 0–4k | | | | 4–8k | | | | 8k-16k | | | |
|---|---|---|---|---|---|---|---|---|---|---|---|---|---|
| | | FoX | FoX-PaTH | PaTH | RoPE | FoX | FoX-PaTH | PaTH | RoPE | FoX | FoX-PaTH | PaTH | RoPE |
| QA | 2wikimqa | 21.0 | 23.7 | **28.7** | 23.9 | 15.3 | **22.5** | 20.8 | 0.9 | **9.4** | 8.4 | 7.3 | 0.1 |
| | hotpotqa | 20.3 | 16.2 | 19.0 | **25.2** | 9.3 | 16.1 | **22.8** | 0.8 | 5.6 | 7.7 | **8.8** | 0.4 |
| | multifieldqa_en | 39.1 | **39.6** | 38.6 | 18.0 | 24.9 | **31.4** | 27.2 | 5.1 | 16.0 | **19.5** | 19.2 | 1.9 |
| | qasper | 22.4 | 24.6 | **25.9** | 15.1 | 14.9 | **19.8** | 16.8 | 1.8 | 7.0 | 10.1 | **10.6** | 1.9 |
| Summarization | multi_news | 9.1 | 6.9 | **12.1** | 10.2 | 7.3 | **9.8** | 9.6 | 3.1 | 6.1 | **8.3** | **8.3** | 1.7 |
| | gov_report | 14.4 | 10.2 | **22.3** | 12.4 | 14.5 | 13.6 | **17.9** | 4.9 | 5.9 | **11.9** | 11.6 | 2.5 |
| Few-shot | trec | 35.0 | 36.7 | **40.0** | 23.3 | 27.5 | 26.3 | **35.0** | 1.2 | 20.6 | **26.3** | 20.0 | 0.0 |
| | triviaqa | 33.2 | 28.9 | **36.0** | 21.8 | 18.2 | 27.6 | **32.0** | 2.8 | 13.7 | **31.6** | 18.4 | 0.4 |
| | samsum | 21.4 | **27.1** | 26.8 | 19.3 | 16.9 | **27.6** | 23.6 | 3.2 | 9.1 | **15.7** | 15.6 | 0.7 |
| Code | lcc | 19.2 | 21.4 | **22.3** | 22.1 | 18.8 | **23.3** | 18.6 | 7.9 | 18.2 | 18.9 | **19.0** | 4.8 |
| | repobench-p | 21.8 | 22.7 | **27.3** | 14.6 | 18.4 | 22.5 | **22.7** | 9.2 | 17.5 | **19.3** | 19.2 | 7.6 |
| **Average** | | 23.4 | 23.5 | **27.2** | 18.7 | 16.9 | 21.9 | **22.5** | 3.7 | 11.7 | **16.1** | 14.4 | 2.0 |

**Table 7:** Performance comparison grouped by task category. Each bolded value indicates the best model score for the respective dataset and length bucket.

