# OpenReview forum: "PaTH Attention: Position Encoding via Accumulating Householder Transformations"
_NeurIPS.cc/2025/Conference — NeurIPS 2025 poster_

### Official Review · Reviewer_6WvM · 2025-06-26

**Clarity:** 3
**Significance:** 3
**Originality:** 3
**Rating:** 5
**Confidence:** 5

**Summary:**

The paper investigates a substitute for the widely-used RoPE positional encoding in language models. Motivated by recent advances in linear RNNs and attention mechanisms, it proposes a multiplicative positional encoding scheme applied between each pair of tokens. It uses the data-dependent, Householder-like matrix with an identity-plus-rank-one structure, where the rank-one components are learnable. Leveraging this structured form, the paper further introduces and implements efficient training kernels that compute the pairwise multiplicative positional encodings efficiently. Experiments are conducted on a 1B-parameter model trained with 50B tokens. Results show that the proposed learnable positional encoding~(PE) outperforms both the classical RoPE and a recently introduced learnable multiplicative PE across synthetic tasks, common-sense reasoning, and long-context benchmarks. The paper also evaluates and compares their performance in length extrapolation scenarios.

**Questions:**

1. I suggest shortening Chapter 3 and including more experimental results on efficiency.

2. I recommend expanding Chapter 2 and providing more exploration on the design of the H matrix.

3. In Table 2, many tasks include both acc and acc_norm as metrics. However, the paper sometimes reports acc, and other times acc_norm. For example, both ARC-e and ARC-c include these two metrics, but the paper reports acc for ARC-e and acc_norm for ARC-c. This inconsistency appears confusing to me.

4. The current design appears quite similar to some recent approaches in linear RNNs or linear attention mechanisms. As noted around lines 70–80, the main difference lies in the use of the softmax function. It would be helpful if the paper could further explore how this softmax non-linearity affects performance, when compared to its linear counterparts.

5. For additional questions, please refer to the weakness section.

**Ethical Concerns:**

["NO or VERY MINOR ethics concerns only"]

**Final Justification:**

I think the main problem of this paper is their efficient implementation. I thought it was due to some clarity problems that there are not many efficiency results in the paper. However, based on the discussion, it seems that this is not the case.

While efficiency has been a main part of the method and the contribution, the training speed of the single-head PaTH is obviously slower than the parallel attention, and decoding has not been implemented yet, which is extremely important, given that the main bottleneck is caused by the attention.

Therefore, I tend to maintain my score.

**Limitations:**

yes

**Paper Formatting Concerns:**

no formatting issue

**Quality:**

3

**Strengths And Weaknesses:**

Strengths
1. The paper addresses the important problem of improving positional encoding in transformer models. Instead of using the fixed RoPE encoding, it introduces a novel learnable multiplicative positional encoding scheme that operates between each token pair.

2. To ensure both expressiveness and computational efficiency, the method employs a Householder-like matrix with an identity-plus-rank-one structure, along with an efficient training pipeline supported by custom Triton kernels.

3. The experiments span a broad range of tasks, demonstrating the method’s effectiveness across classical language modeling benchmarks, long-context tasks, retrieval scenarios, and length extrapolation evaluations.

4. The paper is easy to follow.

Weaknesses
1. The paper lacks sufficient exploration of alternative designs for the matrix H. Although it follows ideas from recent developments in linear RNNs, the presence of the softmax operation in attention could lead to significantly different behavior. Different choices of H could yield not only different performance but also different computational efficiencies. Moreover, the paper does not explain why Householder matrices are particularly suitable or theoretically motivated in this context.

2. The experimental study on efficiency is insufficient. Although the paper takes three pages for its efficient training and inference approach, the empirical evidence is limited to a single figure. This fails to convincingly demonstrate its competitiveness compared to Attention. Assuming Figure 1 represents training efficiency, it would be helpful to include latency results across different model dimensions as well. Additionally, inference-time performance is not reported. Since Eq. (8) suggests extra computations are required for updating the key vectors, reporting decoding latency would help clarify the inference-time cost.

3. The model architecture details are unclear. For example, Fox introduces modifications such as additional normalization layers beyond the standard transformer design. When combining Fox with the proposed PaTH method, the paper does not clarify which architectural components are used in the experiments. This raises questions about the fairness of comparisons to standard attention-based models.

---

> ### Author Response · Authors · 2025-08-01
>
> We thank the reviewer for their constructive feedback.
>
> # Insufficient Exploration of the H Matrix Design
>
> We appreciate the suggestion to clarify our design choices.
>
> Empirically, we found it necessary to set \$\beta \in (0, 2)\$. Fixing \$\beta = 2\$ enforces norm preservation but causes severe numerical instability under BF16, degrading performance and training stability. Despite significant effort, we were unable to stabilize this setting. Using \$\beta \in (0, 1)\$, as in DeltaNet, led to a loss of state-tracking, consistent with findings in *“Unlocking State-Tracking in Linear RNNs Through Negative Eigenvalues.”*
>
> We also found that sharing projection parameters between \$k\$ and \$w\$ (as in DeltaNet and Gated DeltaNet) significantly hurts performance. Unsharing them was beneficial. We plan to include a full ablation study in the revision to support these findings.
>
> # Motivation for Using Householder Matrices
>
> Our primary goal is to improve state tracking, especially for problems like the S5 permutation task, which is NC\$^1\$-complete under AC\$^0\$ reductions. Solving S5 implies the model can simulate a wide class of state-tracking computations.
>
> This motivates us to look for transformations that naturally model compositions of swaps. Householder matrices are appealing in this context: each matrix reflects a vector across a hyperplane and can mimic a pairwise swap under certain conditions, while preserving other coordinates.
>
> By cumulatively composing these Householder matrices (our PaTH scheme), we simulate multiple such swaps—aligning structurally with the S5 group.
>
> This motivation is formalized in Theorem 2.1 (Appendix C), though currently only briefly mentioned in lines 625–626. We agree this should be made clearer earlier in the paper and will revise the introduction and Section 2 accordingly.
>
> # Efficiency Evaluation
>
> #### Triton Kernel Latency Comparison (Forward + Backward, ms)
>
> ```
> | Batch | SeqLen | HeadDim | Parallel Attn | PaTH Attn | Forgetting Attn |
> |-------|--------|---------|----------------|-----------|-----------------|
> | 16    | 2048   | 128     | 5.07           | 17.59     | 8.79            |
> |       |        | 64      | 6.55           | 11.95     | 8.54            |
> | 8     | 4096   | 128     | 8.47           | 21.88     | 11.16           |
> |       |        | 64      | 11.01          | 16.64     | 14.50           |
> | 4     | 8192   | 128     | 15.28          | 30.90     | 20.16           |
> |       |        | 64      | 19.95          | 26.06     | 26.45           |
> | 2     | 16384  | 128     | 28.76          | 48.49     | 37.92           |
> |       |        | 64      | 37.64          | 43.89     | 50.05           |
> ```
> Since FFN dominates training time in practice (esp. in MOE training), the end-to-end training speed difference will be smaller than the kernel-level gap shown here. We are still actively optimizing the kernel for headdim128.
>
> # Inference Latency
>
> Due to time constraints, we did not implement inference-time optimization. We note that decoding latency is highly implementation-dependent, and efficient fused implementations are possible.
>
> In particular, the rank-1 refinement on past key vectors involves a fixed pattern of memory access, making it amenable to a fused kernel that performs Flash-decoding with integrated key refinement. The main overhead is writing updated keys back to HBM, which we believe can be pipelined efficiently. We plan to explore inference latency more thoroughly in future work.
>
> # Model Architecture Details
>
> Thank you for highlighting this. We apologize for the overly brief description. In our experiments, we use the standard LLaMA architecture, rather than the modified “Pro” version in FoX. This was an intentional choice: we aimed to minimize architectural changes, so the effect of replacing RoPE with PaTH can be isolated and fairly compared. All our baselines—RoPE, FoX, and PaTH—use this consistent setting.
>
> The only additional components used by PaTH are those needed to compute `w` and `β`, following *“Unlocking State-Tracking in Linear RNNs Through Negative Eigenvalues”*:
> Linear projection → Short convolution → L2 normalization (per head).
>
> To reduce parameter overhead and ensure fair comparison, we apply low-rank projections, similar in spirit to LoRA.
>
> We will make these architectural choices clearer in the main text.
>
>
>
> # Q: Why report acc for ARC-e but acc\_norm for ARC-c?
>
> We follow the evaluation setup from Mamba (Section E.2.3 of the Mamba paper), which reports `acc` for ARC-e and `acc_norm` for ARC-c.
>
> # Q4: How does the softmax affect performance?
>
> While linear RNNs are efficient, they struggle with retrieval and contextual selectivity. On RULER, removing softmax (reducing to DeltaNet) significantly hurts performance. We will clarify this in the revision and plan to include retrieval comparisons in future work.

---

> > ### Comment · Reviewer_6WvM · 2025-08-04
> > **Response**
> >
> > Thanks for the comments. However, my concerns are not fully addressed.
> >
> > > Our primary goal is to improve state tracking, especially for problems like the S5 permutation.
> >
> > I still don't understand the motivation for improving state tracking in a strong attention model.
> >
> > > Efficiency
> >
> > I initially thought these results might have been overlooked due to issues in writing or clarity. However, it now seems that this is not the case. Although the efficiency section is presented as a major contribution of the paper, the results do not appear convincing, and the inference kernel has not been implemented. For both training and prefilling, attention can still be a bottleneck when the sequence length is long. PaTH seems to double the time with a head dimension of 128 for long sequences. This is only the per-head speed, and PaTH adds additional components such as larger linear projections, convolutions, and normalization layers. Inference latency for attention is especially important, as it remains the primary bottleneck, but this is not implemented.
> >
> > > The only additional components used by PaTH are those needed to compute w and β, following “Unlocking State-Tracking in Linear RNNs Through Negative Eigenvalues”: Linear projection → Short convolution → L2 normalization (per head).
> >
> > Given that PaTH is using the additional components, including short convolution and L2 normalization, and these may also help with the attention, thus, as I stated in my reviews, we are unsure how the positional encoding design proposed in PaTH contributes, especially considering the trade-off of increased latency.

---

> ### Author Response · Authors · 2025-08-05
>
> Thank you for engaging!
>
>
> > I still don't understand the motivation for improving state tracking in a strong attention model.
>
> State tracking remains a challenging and unresolved issue for RoPE-based attention models. A recent benchmark, *LoCoDiff: Natural Long Context Code Bench*, evaluates a key capability for coding agents—tracking the state of edited files—which is a prototypical state tracking task. Performance degrades sharply as context length increases: while some models reach near 100% accuracy on prompts under 5k tokens, accuracy drops significantly by 10k tokens and falls below 50% at 25k tokens. This highlights a fundamental limitation of RoPE-based attention: the number of layers required for effective state tracking grows logarithmically with sequence length, yet the number of layers in practice is fixed.
>
> In contrast, NC¹-complete (under AC⁰ reductions) architectures such as PaTH and DeltaFormer may be capable of solving these state tracking tasks without the need for deeper networks as the input length grows. At a higher level, PaTH may also be better suited for chain-of-thought reasoning tasks, offering improved token efficiency due to its greater expressive power.
>
> While we currently lack empirical results on real-world code tasks, this work focuses on synthetic tasks specifically designed to test state tracking. We view these results as a promising starting point, and acknowledge that extending PaTH to practical scenarios like test-time scaling and code modeling will require significant future work—likely the subject of a follow-up paper.
>
>
>
> > Efficiency
>
> We acknowledge the current limitation of PaTH attention efficiency—its kernel is approximately 2× slower than standard attention (note: this refers to the attention kernel itself, not end-to-end training performance). In future work, we plan to explore *partial PaTH*—applying PaTH to only a subset of the attention dimensions while using NoPE for the rest—similar in spirit to partial RoPE or Multi-Latent Attention, to improve speed without sacrificing too much expressivity. We also aim to rewrite the kernel using more advanced GPU programming languages such as TileLang, since Triton’s current memory management is suboptimal for PaTH, which requires substantial SRAM to efficiently support transition matrices.
>
> Regarding inference latency, we aim to implement PaTH in vLLM for a fair comparison. However, this would require several weeks of engineering effort, so we are unable to provide numbers at this time. We believe that inference speed comparisons using Hugging Face are somewhat unreliable, as HF is significantly slower and inference performance is highly implementation-dependent. That said, if you think an HF-based speed comparison would still be useful, we’re happy to run it and report the results. We also hope that our discussion in Section 3.3 partially addresses your concerns about inference efficiency. Also, as mentioned earlier, partial PaTH appears to be a promising direction for reducing inference complexity further.
>
> Despite this efficiency limitation, we believe it’s important to explore more expressive—even if more expensive—attention mechanisms. While many recent works have focused on hardware-efficient linear attention for better throughput, they often fall short in expressivity and downstream task accuracy. As compute continues to become cheaper, we risk running out of data and model variants that meaningfully push the frontier. Our goal is not to make PaTH faster than RoPE, but to propose a practical, blockwise algorithm that makes PaTH *trainable* on modern hardware—enabling empirical investigation of its unique properties. With the current kernel, for example, we are able to fine-tune a 7B model with head dimension 128 without difficulty (see our response to Reviewer V5VP for results).
>
> We remain optimistic about the direction of developing more expressive and powerful attention mechanisms that remain hardware-aligned—ensuring that wall-clock performance stays affordable even as theoretical complexity grows.
>
> > Given that PaTH is using the additional components, including short convolution and L2 normalization
>
> We did not modify the QKV computation—it remains identical to what is used in LLaMA with RoPE attention. The L2 normalization of $w$ is necessary to ensure the transition matrix has a spectral norm less than 1, which is important for stability. As for the short convolution, it does not have a significant impact on performance. In fact, in our distillation experiment (see our response to Reviewer V5VP), we excluded the short convolution module entirely, and the model still performed well. We plan to clarify these architectural details and conduct ablation studies to make this more transparent.

---

### Official Review · Reviewer_8tf9 · 2025-06-30

**Clarity:** 2
**Significance:** 3
**Originality:** 3
**Rating:** 4
**Confidence:** 2

**Summary:**

This paper introduces a new method called PATH, which is a flexible data-dependent position encoding method that employs a Householder-like transformation. They provide  theoretical analyses to understand its method.  The paper has also achieved a hardware-efficient implementation of this method. Through multiple experiments, the effectiveness of this method has been demonstrated.

**Questions:**

1. Comparative Experiment with NoPE ( The impact of positional encoding on length generalization in transformers)

    In the NoPE paper, the authors demonstrated that even without positional encoding, transformers can still perceive position information. In this work, the authors have introduced a data-dependent position encoding method by incorporating a function related to the input. Given that neural networks are capable of simulating such functions, it would be valuable to include a comparative experiment with NoPE to highlight the effectiveness of this design. This comparison would provide clearer insights into the benefits of the proposed data-dependent position encoding.

2. Discrepancy and Concerns in the Flip-flop Experiment

   In the Flip-flop experiment, it has been theoretically proven that two layers can solve the task. However, the actual implementation only utilizes one layer. Could the authors elaborate on the differences between the theoretical and practical outcomes? Additionally, there are some concerns regarding the experimental setup. Since the experiment only employs one layer and the PATH also includes the input function, especially with a limited vocabulary of only 5, it seems that the input function might play a more significant role. This could potentially skew the results and make the comparison unfair. It would be more convincing if the authors could increase the number of layers and redo the experiment to provide a more comprehensive evaluation.

**Ethical Concerns:**

["NO or VERY MINOR ethics concerns only"]

**Limitations:**

yes

**Quality:**

3

**Strengths And Weaknesses:**

Strengths：

1. The authors introduce a novel data-dependent position encoding method and provide a hardware-efficient implementation. This represents an innovation in the field.
2. The authors offer theoretical proof that the newly designed method can effectively address the state tracking issue. This has enlightening implications for enhancing the capabilities of existing models.
3. Through multiple experiments, the authors have demonstrated the effectiveness of the proposed method.

Weaknesses:

The actual implementation of the algorithm is not very clear. In line 69, the actual formula is given, and then it involves using the UT form (line 121) and hardware-efficient implementation to accumulate P. Since the method is data-dependent and involves a function related to the input, it would be beneficial if the authors could provide a more detailed algorithm flow. Specifically, how is the function integrated into U or P, and what is the actual workflow?

---

> ### Author Response · Authors · 2025-08-01
> **Rebuttal**
>
> Thank you for the constructive feedback.
> #  Implementation detail
>  We aim to improve the clarity of the algorithm flow in the next iteration. For now, we provide a PyTorch reference implementation for illustration. (For simplicity, we omit the online softmax trick and explicitly construct the attention matrix `A`; in our actual Triton implementation, `A` is never materialized and online softmax is used.) Inputs are shaped `[B, H, L, D]` for `q`, `k`, `v`, `w`, and `[B, H, L]` for `beta`.
> ```
> import torch
> from einops import rearrange
> def naive_path_attn(q, k, v, w, beta, scale, BT=64):
>     b, h, l, d = q.shape
>     wb = w * beta[..., None]
>     q, k, w, wb = [rearrange(x, 'b h (n c) d -> b h n c d', c=BT) for x in (q, k, w, wb)]
>     m = torch.triu(torch.ones(BT, BT, dtype=torch.bool))
>     T = -(wb @ w.transpose(-1, -2)).masked_fill(m, 0)
>     # forward substituition
>     for i in range(1, BT):
>         T[..., i, :i] += (T[..., i, :, None] * T[..., :, :i]).sum(-2)
>     T = T + torch.eye(BT)
>     Twbk = T @ (wb @ k.transpose(-1, -2)).masked_fill(m, 0)
>     qw = (q @ w.transpose(-1, -2)).tril()
>     Twb = T @ wb
>     Al = (q @ k.transpose(-1, -2)).tril() - qw @ Twbk
>     q = q - qw @ Twb
>     k = k - Twbk.transpose(-1, -2) @ w
>     H = w.transpose(-1, -2) @ Twb
>     q, k = [rearrange(x, 'b h n c d -> b h (n c) d') for x in (q, k)]
>     A = torch.zeros(b, h, l, l)
>     # FlashAttention-2 style blocking
>     for i in range(0, l, BT):
>         qi = q[:, :, i:i+BT]
>         for j in range(i-BT, -1, -BT):
>             kj = k[:, :, j:j+BT]
>             A[:, :, i:i+BT, j:j+BT] = qi @ kj.transpose(-1, -2)
>             qi = qi - qi @ H[:, :, j//BT]
>     for i in range(l//BT):
>         A[:, :, i*BT:(i+1)*BT, i*BT:(i+1)*BT] = Al[:, :, i]
>     A = A.masked_fill_(~torch.tril(torch.ones(l, l, dtype=torch.bool)), float('-inf'))
>     return (A * scale).softmax(-1) @ v
> ```
> # Regarding NoPE
> Thank you for the concern. We view NoPE as orthogonal to our work, and also length extrapolation is not our primary focus but rather a byproduct of our data-dependent encoding. We didn’t compare with NoPE due to time constraints. Moreover, we want to highlight that NoPE shares RoPE’s theoretical limitations (both in TC⁰), while PaTH goes beyond TC⁰ to support parallelizable state tracking.
>
> # Regarding FFLM
> Regarding the discrepancy between theory and experiment: this stems from the fact that, in our implementation, we apply a short convolution layer after `w_proj` to compute `w`. In the theoretical construction, however, w is used directly without convolution, requiring the first layer to perform explicit token shifting—hence, two layers are needed. With short convolution, the token shift is effectively handled by the convolution itself, so theoretically only one layer is sufficient to solve FLIP-FLOP, consistent with our experimental results. This is analogous to the case of induction heads: a single layer cannot implement the induction mechanism, but two layers can. However, if the short convolution is applied to the key/value projections, a single layer can implement the induction head, as shown in [1, 2].
>
> We agree it’s unfair to compare shortconv-enhanced PaTH to others—thank you for pointing this out. As suggested, we report results across layer counts for fairness (best from LR ∈ {3e-4, 6e-4, 1e-3}):
>
> | Model          | Layers | Head Dim | IID Error | OOD Sparse Error | OOD Dense Error |
> | -------------- | ------ | -------- | -------- | ---------- | --------- |
> | RoPE           | 8      | 64       | 0%       | 5%         | 0%        |
> | RoPE           | 4      | 64       | 0%      | 5%         | 0%        |
> | RoPE           | 2      | 64       | 0%       | 5%         | 0%        |
> | RoPE           | 1      | 64       | 12%      | 43%        | 0.1%      |
> | FoX            | 1      | 64       | 0.01%    | 23%        | 0%        |
> | FoX            | 2      | 64       | 0%       | 1%         | 0%        |
> | FoX            | 4      | 64       | 0%       | 1%         | 0%        |
> | FoX            | 8      | 64       | 0%       | 1%         | 0%        |
> | Stick-Breaking | 2      | 64       | 0%       | 0%         | 0%        |
> | Stick-Breaking | 1      | 64       | 10%      | 38%        | 0.001%    |
> | PaTH           | 1      | 64       | 0%       | 0%         | 0%        |
>
> Small OOD errors persist for RoPE/FoX even when stacking multiple layers (see FFLM paper appendix for theoretical insight) .  Though stick-Breaking attention solves FFLM with 2 layers, it underperforms PaTH in A5 word problem and MQRAR-N tasks. Overall state tracking ability:  RoPE ~= FoX < SBA < PaTH.
>
> [1] KV Shifting Attention Enhances Language Modeling
>
> [2] Physics of language model 4.1

---

> > ### Comment · Reviewer_8tf9 · 2025-08-06
> >
> > Thank you for the rebuttal — it has addressed most of my concerns.
> >
> > I still have a few questions regarding NoPE.
> >
> > Although NoPE involves length extrapolation, its main point seems to be that the model can still work without any additional positional encoding design. Could you clarify why you claim that **NoPE shares RoPE’s theoretical limitations  (both in TC⁰)**?

---

> > > ### Author Response · Authors · 2025-08-06
> > >
> > > Thank you for the response.
> > >
> > > Regarding the question about RoPE and NoPE, it has been shown that both RoPE and NoPE belong to the circuit complexity class TC0 in [1,2]. Intuitively, this shows that NoPE and RoPE-based architectures have a similar representation power upper bound.
> > >
> > >
> > > [1] Transformers in Uniform TC^0
> > >
> > > [2] Circuit Complexity Bounds for RoPE-based Transformer Architecture

---

### Official Review · Reviewer_V5VP · 2025-07-02

**Clarity:** 4
**Significance:** 4
**Originality:** 4
**Rating:** 5
**Confidence:** 5

**Summary:**

PaTH attention proposes a novel way to encode the positional difference between two tokens in the sequence in a data-dependent manner, not like RoPE (data-independent). This allows us to solve the problems that are not possible to be solved or very hard in RoPE-based attention networks, such as flip-flop language models.

**Questions:**

- Can you release the code during the rebuttal period? I wanna see actual implementation, and how it is complicated as the paper claimed in the limitation section.

**Ethical Concerns:**

["NO or VERY MINOR ethics concerns only"]

**Final Justification:**

All my concerns are resolved, and I believe this paper is strong.

**Limitations:**

I think the limitation section makes sense and is explicit. However, I want to see more fundamental methodology level limitation, rather than simply claiming common limitations of academic research (limited computing resources and time, limited engineering powers). For example, I want to see how this method is compatible with Flash Attention 3 style optimization (async matmuls, tensor memory accessors, specialized warps, and more)

**Paper Formatting Concerns:**

I like the paper formatting, but if a main figure that explains how the PaTH works at a high level would be helpful.

**Quality:**

4

**Strengths And Weaknesses:**

# Strength
- Their proposed PaTH attention is data-dependent, solves problems of existing softmax attention with RoPE.
- They also showed a hardware-efficient kernel for inference and training the PaTH attention, which is a very impressive part.

# Weakness
- This method is not directly compatible with an ordinary RoPE-based attention model, so we cannot apply PaTH attention in a plug-and-play manner. I think at least we should try to transfer training (like this: https://arxiv.org/html/2310.01777v2) from RoPE to PaTH on already released models such as Qwen3. I believe swapping existing softmax attention to PaTH and training with knowledge distillation might make it possible to run PaTH on the existing state-of-the-art models. This would be cheaper to train, and make your method way more impactful. I strongly suggest doing this, even after this submission or paper decisions.
- Their methodology section is a little bit hard to follow, because every explanation is in text and equations. I think it would be great if we had some figures about methodology and algorithm, like previous papers (FA: https://arxiv.org/abs/2205.14135, Figure 1 Center, LA: https://arxiv.org/pdf/2401.04658, Figure 2).
- The training curve is missing. I believe the pretraining loss curve is essential information for evaluating whether the training is complete and the proposed method is fully converged compared to the baseline (softmax attention). It might be too small a corpus to train the softmax attention-based Transformer with 50B tokens. So, I think the loss curve might be helpful to convince readers that this method is training very well, compared to existing baselines.
- Lacks of open implementation. The authors did not submit their code.

---

> ### Author Response · Authors · 2025-07-31
> **Rebuttal**
>
> Thanks for the review. Due to restrictions imposed by the NeurIPS committee, we are unable to provide new figures or code here. However, we commit to open-sourcing the highly optimized Triton implementation.
>
> # Replacing RoPE with PaTH from an Existing Checkpoint
>
> Thank you for the insightful suggestion regarding distillation. We followed the Rapid Attention Distillation pipeline [1] to distill a teacher model using RoPE into a student model with PaTH attention.
>
> We initialize PaTH using the teacher model’s weights, including all MLP and attention layer parameters. Since PaTH introduces an additional `w_proj` parameter, we initialize it with the weights from `k_proj`. Both `k_proj` and `w_proj` are then trained independently.
>
> Following [1], we use the `DCLM ` dataset for training. The distillation process involves two stages:
>
> * **Stage 1:** Align the outputs of each attention layer between the teacher (RoPE) and student (PaTH) using MSE loss. Only the attention layers are trained, while MLP layers remain frozen. This stage is run for 100M tokens with a context length of 512.
> * **Stage 2:** Perform standard knowledge distillation (using KL divergence on the vocab logits)  with full finetuning across all layers. This stage is run for 3B tokens with a context length of 4K.
>
> We summarize selected results below.
>
> The teacher model is `Qwen2.5-7B-Instruct`. We use OpenCompass’s recommended settings for evaluating Qwen-Instruct series models (please refer to the corresponding markdown files in their repository for task-specific configurations).
>
>
> **Results using OpenCompass’s recommended configs:**
>
> | Task          | Teacher (RoPE) | Student (Distilled PaTH) |
> | ------------- | -------------- | ------------------------ |
> | MMLU          | 74.21          | 73.28                    |
> | Hellaswag     | 85.20          | 84.83                    |
> | Winogrande    | 71.51          | 68.90                    |
> | GPQA_Diamond | 33.33          | 34.34                    |
> | TheoremQA |  18.12 | 21.88 |
> | GSM-8K        | 80.29          | 80.67                    |
> | MATH          | 69.10          | 65.38                    |
> | HumanEval     | 82.32          | 77.44                    |
> | MBPP          | 74.71          | 75.10                    |
>
> We observe that the distilled student PaTH model largely recovers the performance of the teacher on these challenging tasks, and in some cases even outperforms it (e.g., GSM-8K, MBPP, TheoremQA, GPQA).
>
> We believe PaTH has strong potential in coding, reasoning, and math tasks. As a next step, we plan to apply distillation followed by supervised fine-tuning (SFT) on coding-specialized models such as Qwen2.5-Coder and Qwen2.5-Math, using a mix of code and math data. Evaluating the reasoning capabilities of these distilled models is also of great interest, which we leave for future work.
>
>
> We also evaluate on RULER using `lm-evaluation-harness`:
>
> | Task       | Teacher (RoPE) | Student (Distilled PaTH) |
> | ---------- | -------------- | ------------------------ |
> | RULER (4K) | 94.37          | 93.24                    |
>
> We find that PaTH largely recovers the teacher model’s performance at the training length of 4K. However, performance degrades when evaluating beyond this length, indicating that the student model does not fully retain the teacher’s long-context capability. This suggests the need for further sequence length extension experiments, which we leave for future work.
>
> [1] RADLADS: Rapid Attention Distillation to Linear Attention Decoders at Scale

---

> > ### Comment · Reviewer_V5VP · 2025-08-05
> >
> > Thank you for your insightful response.
> >
> > > Due to restrictions imposed by the NeurIPS committee, we are unable to provide new figures or code here.
> >
> > Yes, it is quite disappointing that we cannot upload the PDF to the rebuttal. Can you at least post the loss graph with tables?
> >
> > > We observe that the distilled student PaTH model largely recovers the performance of the teacher on these challenging tasks
> >
> > It is pretty cool to see that PaTH (which is completely different from RoPE) could work with KD. I suggest fine-tuning with more tokens (over 100B) to show better performance. I hope to see that updated model weight release after the decision!
> >
> > ---
> >
> > Since I think all my concerns are resolved, especially about the adaptation performance using knowledge distillation, I will raise the score from 4 to 5.
> >
> > Thanks for the insightful paper!

---

> ### Author Response · Authors · 2025-08-05
>
> Thank you for raising the score! We plan to conduct more distillation and continual pretraining experiments over the next month and will make the weights publicly available.
>
> >  Can you at least post the loss graph with tables?
>
> We used `wandb` to log the loss and applied EMA smoothing with a coefficient of 0.6. Below is the reported loss at different training steps (We trained for 23,840 steps with a batch size of 2M tokens, amounting to approximately 50B tokens in total)
>
> | Step   | RoPE  | FoX   | PaTH  | PaTH-FoX |
> | ------ | ----- | ----- | ----- | -------- |
> | 5,000  | 2.504 | 2.493 | 2.483 | 2.450    |
> | 10,000 | 2.376 | 2.367 | 2.352 | 2.328    |
> | 15,000 | 2.314 | 2.305 | 2.288 | 2.269    |
> | 23,840  | 2.236 | 2.272 | 2.210 | 2.192    |
>
> We observe a consistent trend: `PaTH` achieves **\~0.02** lower loss than the `RoPE` baseline, while `PaTH-FoX` improves by **\~0.04**.

---

### Official Review · Reviewer_UFkx · 2025-07-03

**Clarity:** 4
**Significance:** 4
**Originality:** 4
**Rating:** 5
**Confidence:** 4

**Summary:**

The paper presents a novel position encoding method for transformers that uses data-dependent, accumulated Householder transformations to overcome the expressivity limitations of existing methods like RoPE. Good performance is demonstrated on synthetic benchmarks and real-world language tasks. The method can be efficiently trained at scale using a custom FlashAttention-style algorithm. Ideas on representative power are leveraged from the RNN literature (RWKV-7 and related).

**Questions:**

- any insights how your method changes attention patterns?
- can you give more clarity/evidence why the data-dependency is a benefit, compared to traditional pos embeddings which are added to the data  (values) also?
- (optional) as you inherit many ideas from RWKV-7: would it make sense to ask if similar behaviors (lost in the middle) are present in your transformer compared to RNNs, empirically?
- any idea if the method could be added to existing LLMs with slight post-training, or needs to be there from scratch? (one can always dream)
- detail: can you clarify if BoD tokens or meta-tokens were used (related to attention sinks)

**Ethical Concerns:**

["NO or VERY MINOR ethics concerns only"]

**Final Justification:**

I'm fine with the outcome of the rebuttal, and remain on the accept side (also given the other positive assessments by reviewers)

**Limitations:**

- details: clarify which mistral tokenizer

**Paper Formatting Concerns:**

-

**Quality:**

4

**Strengths And Weaknesses:**

Strengths:
- I'm impressed by the overall integration of conceptual results with real-world impact, showing benefits on relevant benchmarks in language modeling and long-context tasks at scale, while also showing clock-time speed gains.
- Elegant idea with the identity plus rank-1 structures
- Convincing mix of evaluations

Weaknesses
- (minor) A bit unclear on the practical implications of NC^1 vs. TC^0 (though I do appreciate the synthetic task and proofs)
- (minor) Nope is not included as another baseline? (Seems to re-gain popularity these days, alone, and also with llama-4 / gemma-style alternating local/global attention, but that's probably orthogonal?)

---

> ### Author Response · Authors · 2025-08-01
>
> We thank the reviewer for their constructive feedback.
>
> # Re. Weaknesses
>
> ## Practical implications of NC¹ vs. TC⁰
>
> We believe NC¹ architectures may benefit tasks that require strong state tracking, such as code modeling & reasoning tasks. A weak empirical signal supporting this is shown in Figure 3, where PaTH achieves lower perplexity on the CodeParrot corpus. We plan to run more targeted code modeling experiments and explore distillation into coding models to further assess whether PaTH offers consistent advantages in this domain.
>
> ## NoPE not included as a baseline
>
> The primary goal of this work is to improve state tracking rather than length extrapolation. That said, we agree that NoPE is orthogonal and potentially complementary. For example, one could build a hybrid approach similar to MLA—using NoPE across most of the hidden dimensions and applying PaTH only to a subset. Exploring such partially structured designs is an interesting direction for future work.
>
>
>
> # Re. Questions
>
> ## Q2: Why is data-dependency a benefit compared to additive positional embeddings?
>
> We refer the reviewer to Section 5.2 of *“The Illusion of State in State Space Models”*, which discusses how data-dependent transition dynamics can significantly enhance expressivity. Without such data-dependence, PaTH cannot perform proper state tracking—which is the core motivation of this work.
>
> ## Q4: Can PaTH be integrated into existing LLMs with post-training?
>
> Yes—PaTH can be initialized from existing pretrained Transformer checkpoints and post-trained. We demonstrate this via a distillation experiment in our response to Reviewer V5VP, showing the feasibility of retrofitting PaTH onto pretrained models.
>
> ## Q1, Q3, Q5: Attention patterns, lost-in-the-middle, and special tokens
>
> We use a `<bos>` token at the beginning of each document but do not employ additional meta-tokens. We are interested in further analyzing PaTH’s attention behavior—e.g., whether it exhibits attention sinks or "lost-in-the-middle" effects—in future iterations.

---

### Decision · Program_Chairs · 2025-09-17

**Decision:**

Accept (poster)

**Comment:**

The paper introduces a variant of attention, PaTH, which uses a cumulative Householder transformations along with a new data-dependent position encoding scheme. Author provides theoretical expressivity results as well as efficient implementation in FlashAttention-style kernel. Empirical studies show gains on synthetic reasoning, some LM benchmarks, and long-context generalization. All the reviewers were mostly positive about the submission, yet raised important concerns on clarity and details of the algorithm, missing NoPE baseline comparisons, and practical applicability (e.g. whether PaTH could retrofit existing pretrained models). We thank the authors and reviewers for engaging during the rebuttal period to improve the paper. The authors provided a PyTorch reference implementation, clarified the flip-flop setup, and importantly demonstrated knowledge distillation from RoPE to PaTH on Qwen models, showing PaTH largely recovers or improves performance. Based on this one reviewer raised their score, and the others remained positive. While the work would benefit from broader baselines and larger-scale validation with open-sourced code, the submission does provide the community with an attention variant with increased theoretical expressivity and strong experimental results.